# In vivo antimicrobial activity of engineered mesoporous silica nanoparticles targeting intracellular mycobacteria

John Jairo Aguilera-Correa[1,2,5], Yara Tasrini [1,5], Miguel Gisbert-Garzarán [3,5], Aude Boulay[1], Tamara Carvalho [1], Fabien P. Blanchet [1,4], María Vallet-Regí[3] & Laurent Kremer [1,4] ✉

Treatments of *Mycobacterium marinum*, a common non-tuberculous mycobacterium associated with cutaneous infections are very challenging, emphasizing the development of new therapeutic approaches. Here we report the functionalization of mesoporous silica nanoparticles (MSN) with a series of triphenylphosphonium (TPP) substituents, which endowed them with affinity towards the surface of *M. marinum* in vitro, as well as within infected THP-1 cells. The presence of these nanoparticles at the bacterial surface prevents their uptake by human macrophages and dendritic cells. When loaded with doxycycline, the nanosystem exerts a potent anti-bacterial effect in planktonic cultures, biofilms, and in *M. marinum*-infected macrophages. Strikingly, in the *M. marinum*/zebrafish infection model, the doxycycline-loaded nanoparticles are associated with a pronounced decrease in the bacterial burden and a high embryo survival rate. These results disclose the proposed MSN nanosystems as a promising alternative for the treatment of *M. marinum* infection and, presumably, against a broader range of mycobacterial infections.

*Mycobacterium marinum* (*Mmar*) is a photochromogenic, slow-growing, non-tuberculous mycobacterium that can cause infection in humans if the skin barrier is compromised and comes into contact with contaminated water[1], or if directly inoculated by a contaminated animal[2]. Current incidence of *Mmar* infection ranges between 0.04 and 0.13 per 100,000 person-year in Europe[1,3], reaching up to 0.27 per 100,000 person-year in United States[4]. After crossing the skin barrier, this mycobacterium provokes a superficial skin infection characterized by a unique papular and/or ulcerating subcutaneous nodule or granuloma in two-thirds of cases[3]. Granulomas are encircled by a lymphocyte ring, and contain a compact, well-organized collection of immune cells, namely foamy macrophages, differentiated macrophages originating from blood, and multinucleated giant cells[5]. *Mmar* lesions are mainly restricted to cooler areas of the body (e.g., hands

and feet), making this bacterium among the most frequent mycobacterial species causing cutaneous infections[6]. These nodular cutaneous lesions often progress to tenosynovitis, arthritis, and osteomyelitis[3,7]. In up to one quarter of the cases, *Mmar* infection adopts a sporotrichoid form, where infection spreads throughout the lymphatic vessels to the regional lymph nodes, resulting in multiple sporotrichosis-like nodules[4].

Current treatment options for superficial *Mmar* cutaneous infections include doxycycline, minocycline, clarithromycin or trimethoprim-sulfamethoxazole as monotherapy[8]. Nonetheless, the mean duration of those treatments is estimated to $25 \pm 14$ weeks plus $6 \pm 3$ weeks after clinical cure[9]. The clinical benefits of continuous antibiotic therapy decreases with time, while the emergence of side effects, superinfections, and antibiotic resistance increases[10].

[1]Centre National de la Recherche Scientifique UMR 9004, Institut de Recherche en Infectiologie de Montpellier (IRIM), Université de Montpellier, Montpellier, France. [2]CIBERINFEC-CIBER de Enfermedades Infecciosas, Madrid, Spain. [3]Departamento de Química en Ciencias Farmacéuticas, Universidad Complutense de Madrid, Instituto de Investigación Sanitaria Hospital 12 de Octubre i+12, Madrid, Spain. [4]INSERM, IRIM, 34293 Montpellier, France. [5]These authors contributed equally: John Jairo Aguilera-Correa, Yara Tasrini, Miguel Gisbert-Garzarán. ✉e-mail: laurent.kremer@irim.cnrs.fr

Considering the antibiotic mantra "shorter is better"[11], a curtailed but equally effective *Mmar* treatment would be beneficial for the patient and reduce the socioeconomic burden ascribed to this disease.

The application of nanotechnology to medicine, so-called nanomedicine, has attracted much attention from researchers and clinicians. Despite *ca.* 90 nanomedicines approved by the Food and Drug Administration (FDA) with many currently tested in clinical trials, only one FDA-approved, mycobacterium-related nanomedicine is currently available[12]. Preclinical research involving nanoparticle-mediated treatment of *Mmar*[13–16] demonstrated a reduced bacterial burden in the zebrafish model of infection. *Mmar* infection spreads *via* the lymphatic nodules following uptake by dendritic cells (DCs)[17], thereby disseminating the infection throughout the organism. Hence, engineering nanoparticles capable of hampering the mycobacterial uptake by DCs would represent a promising approach to circumvent the potential invasion of healthy tissues.

Mesoporous silica nanoparticles (MSNs) were employed as nanocarriers owing to their outstanding physicochemical features, biocompatibility, and ease of functionalization[18–20]. Only a few examples of mycobactericidal MSNs are available to date, displaying nanocarriers capable of delivering a range of payloads from standard antimycobacterial drugs to novel antimycobacterial peptides[21–26]. Nonetheless, how these nanoparticles behave within infected cells remains largely unknown. In this regard, dendritic MSNs that could efficiently eliminate intracellular mycobacteria but failed to accumulate around the mycobacterial phagosomes were reported[27]. Triphenylphosphine (TPP) is a lipophilic cation that has long been employed as endosomal escape agent[28] and shown to interact with negatively charged mycobacterial membranes[29]. Moreover, according to current knowledge, increasing the hydrophobicity might lead to increased interactions with mycobacterial membranes[30]. Following this rationale, we functionalized the surface of MSNs with a series of (TPP) derivatives as well as series of alkyl 5-aminovaleric acid (AVA) units between the MSN surface and the TPP moiety to increase the lipophilic character of the nanosystem.

In this work, these TPP-functionalized MSNs were screened for mycobacterial and intracellular affinity, and the leading candidate investigated as a potential carrier of antimycobacterial drugs against *Mmar*. Finally, a drug-loaded TPP-functionalized MSN was assessed for its activity in *Mmar*-infected macrophages, exhibiting strong efficacy in eliminating the infection in the zebrafish infection model.

## Results

### Nanoparticle characterization
MSNs were synthesized using TEOS as a silica source and a cationic surfactant as structure directing agent. Following amino surface modification, the nanoparticles were functionalized with a series of AVA$_n$-TPP units (n = 0–2) using carbodiimide chemistry. Fmoc-protected AVA was employed to avoid undesired side reactions. The carboxylic acid of each AVA unit was first activated and subsequently grafted to the available amino groups on the surface. Then, the samples were subjected to Fmoc deprotection to allow for the next grafting step and the process was repeated until all AVA units and TPP were incorporated (Fig. 1a(1)). FTIR spectroscopy confirmed the proper formation of the silica backbone (Fig. 1b). The appearance of C-H vibration bands at *ca.* 2900 cm$^{-1}$ ascribed to the alkyl chain of APTES confirmed the amino modification of the surface. Subsequent grafting of either AVA units or TPP generated new vibrations bands ascribed to the CH$_2$ groups (*ca.* 2900 cm) from the alkyl chains and the amide groups formed after the carbodiimide-mediated coupling (*ca.* 1500–1650 cm$^{-1}$). The as-synthesized MSNs were visually inspected by transmission electron microscopy, confirming the production of homogeneous, quasi-spherical nanoparticles of approximately 200 nm in size (Fig. 1c). The mean size was evaluated after each functionalization step in both water and Dulbecco's Modified Eagle

Medium (DMEM) supplemented with 10% Fetal bovine serum (FBS) to mimic a more complex biological environment (Fig. 1d). Overall, all biologically tested candidates (MSN and TPP-containing nanoparticles) were shown to be stable in both conditions, observing the lowest value for MSN-AVA-TPP in both conditions. The different functionalization steps were also followed through zeta potential measurements, showing sequential variations following each reaction step (Fig. 1e). The mean size value observed for the intermediate material MSN-NH$_2$ in water was ascribed to the low zeta potential value that minimized electrostatic repulsions between nanoparticles, causing them to aggregate. Finally, thermogravimetric analysis showed overall decrease in weight loss after grafting the AVA units and/or TPP (Fig. 1f). The affinity of these functionalized MSNs towards *Mmar*, macrophages, as well as their antimycobacterial properties were subsequently determined in a set of biological assays (Fig. 1a (2-3)).

### TPP confers MSN affinity towards mycobacteria
The TPP-functionalized nanoparticles were initially screened to check their affinity towards the mycobacterial membrane[29]. For that purpose, *Mmar* expressing red fluorescent mScarlet (mScarlet-*Mmar*) was incubated with the different FITC-labeled nanoparticles and subjected to flow cytometry after a 5-min incubation (Supplementary Fig. 1 for gating strategy). While the non-functionalized MSNs barely interacted with the bacteria (9.8%), TPP functionalization strongly increased the affinity towards the mycobacterial surface up to 83.1% (Fig. 2a, b). Incorporating an AVA unit (MSN-AVA-TPP) did not have a significant effect on the interaction compared to MSN-TPP whereas addition of a second AVA unit (MSN-AVA$_2$-TPP) appeared slightly detrimental to the system as compared to MSN-AVA-TPP (Fig. 2a, b). The same affinity pattern was observed with other slow- and rapid-growing mycobacterial species (Supplementary Table 1), confirming the specific affinity of the functionalized nanosystems to the mycobacterial membrane (Fig. 2a, b). Of note, amongst the 6 tested mycobacterial species, the highest affinity was observed with *Mmar*. Scanning electron microscopy confirmed the negligible interaction between MSN (Fig. 2c) (white arrows) and *Mmar* and the massive accumulation for MSN-TPP and MSN-AVA-TPP surrounding the bacilli, further validating their affinity towards the bacterial surface (Fig. 2c). The affinity assay was also performed in the presence of 10% human serum (HS) to evaluate how the TPP-functionalized nanoparticles would interact with bacteria in a more realistic scenario. Even though both MSN-TPP and MSN-AVA-TPP showed decreased affinity, MSN-AVA-TPP performed better in the presence of serum, suggesting that introducing the AVA unit would be beneficial (Supplementary Fig. 2).

### TPP-functionalized MSNs show increased uptake by THP-1 cells
Having demonstrated the affinity of the nanocarriers towards the bacterial surface, we tested their ability to be taken up by cells. First, THP-1 macrophage-like cells were incubated with the different FITC-labeled nanoparticles, demonstrating cellular biocompatibility up to 500 μg/mL even up to 72 h post-incubation (Supplementary Fig. 3a, b). Based on these results, a concentration of 200 μg/mL was selected for subsequent experiments. THP-1 cells were infected with mScarlet-*Mmar* using a multiplicity of infection (MOI) of 10 for 2 h prior to treatment with the different nanosystems for an additional 2 h (Fig. 3a). Automated image analysis of full wells was set up to quantify the affinity and internalization of the nanosystems towards the cells by segmenting cells, bacteria and nanoparticles (Supplementary Fig. 4a). The number of segmented cells was similar in all the conditions and replicates (*ca.* 5 × 10$^3$ cells/well) (Supplementary Fig. 4b). As with bacteria, all TPP-containing nanoparticles showed increased uptake by the THP-1 cells, compared to MSN (Fig. 3b). Of note, undergoing mycobacterial infection did not alter the phagocytic capacity of the cells, as we did not observe significant differences in the uptake of the nanoparticles between the non-infected and infected cells (Fig. 3b).

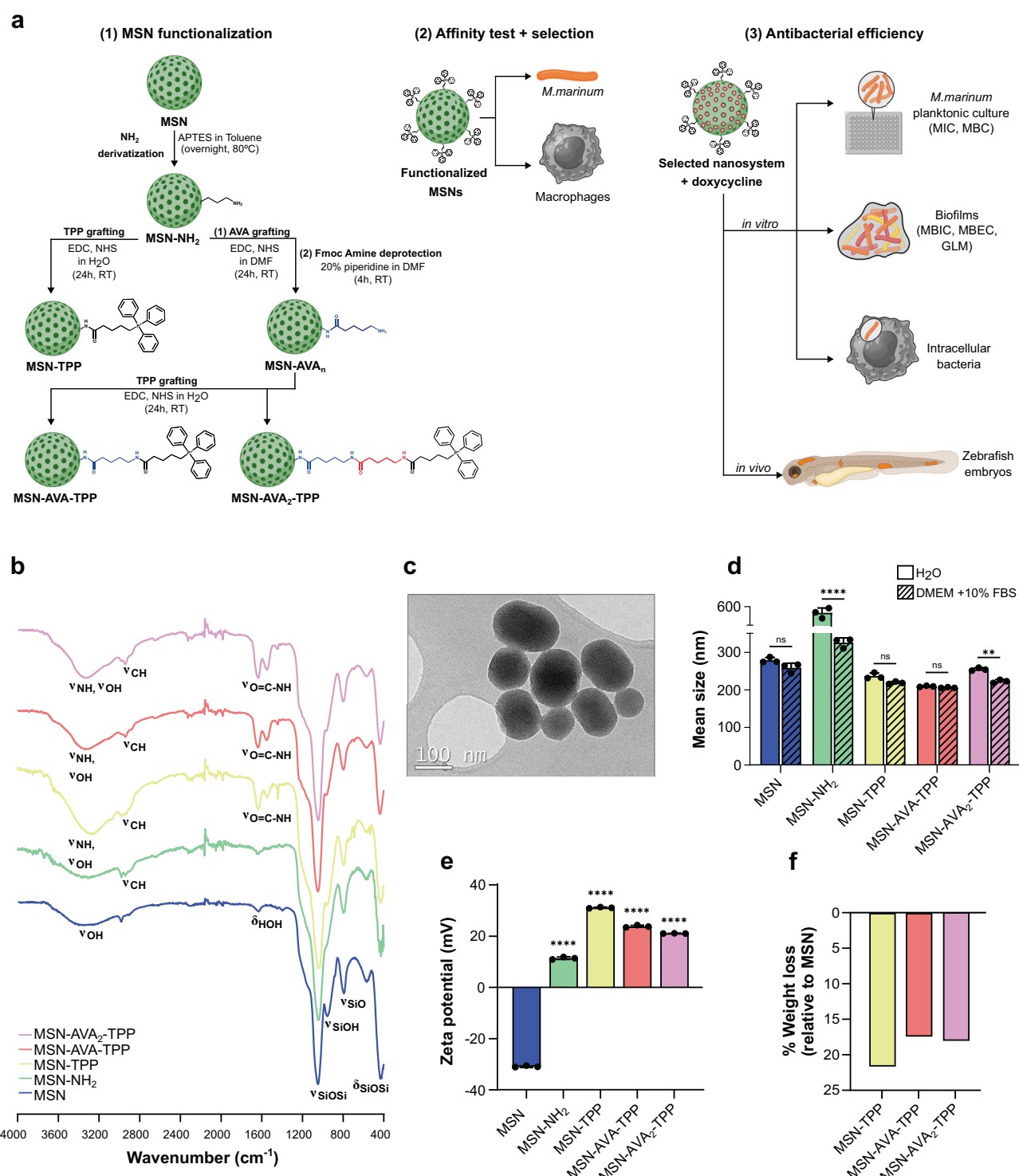

**Fig. 1 | Synthesis and physicochemical characterization of the different nanosystems. a** (1) Schematic representation of the nanosystem functionalization and chemical structure of each AVA$_n$-TPP, (n = 0, 1, or 2). (2-3) Rational and assay schematic of our study. **b** FTIR spectroscopy of the different nanocarriers. **c** Representative Transmission Electron Microscopy (TEM) micrograph of the as-synthesized MSN. Scale bar: 100 nm. Experiment was performed 3 independent times. **d** Mean size of the different nanomaterials in water (plain colors) and Dulbecco's Modified Eagle Medium (DMEM) containing 10% FBS (dashed colors). Each dot corresponds to the mean value of a measurement run. Data are shown as mean ± SD (n = 3). Comparisons between the two media were done using two-sided multiple unpaired t tests with Holm-Šídák's multiple comparisons test. Adjusted pvalues: ns p = 0.073 (MSN), ****p = 0.000095 (NH$_2$), ns p = 0.068 (TPP), ns p = 0.068 (AVA), **p = 0.002 (AVA$_2$). **e** Zeta potential values after each functionalization step. Each dot corresponds to the mean value of a measurement run. Three independent runs were performed. Data are shown as mean ± SD (n = 3). Comparisons between functionalized nanosystems and control MSN were done using ordinary one-way ANOVA with Šídák's multiple comparisons test. Adjusted p values: ****p < 0.0001. **f** Percentage of weight loss of the functionalized nanosystems vs MSN. Source data are provided as a SourceData file. Created in BioRender. Kremer, L. (2025) https://BioRender.com/p3hsx73.

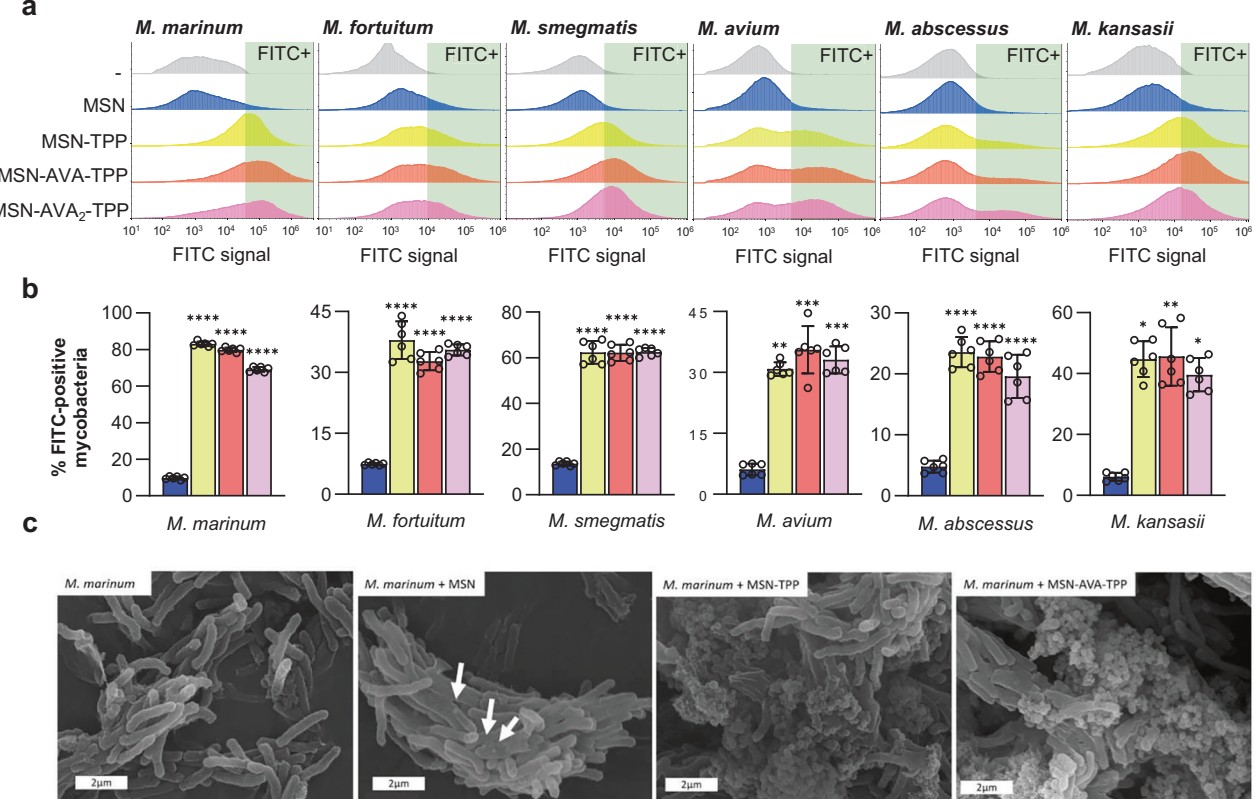

**Fig. 2 | Affinity of the different nanosystems towards different mycobacteria.** **a** Representative flow cytometry analysis of different strains of mycobacteria incubated with PBS (gray), or 200 μg/mL of MSN (blue), MSN-TPP (yellow), MSN-AVA-TPP (orange) and MSN-AVA₂-TPP (pink). Histograms were generated on the bacterial population expressing mScarlet, tdTomato or mCherry, detailed in the gating strategy (Supplementary Fig. 1). **b** Percentages of FITC-positive myco-bacteria incubated with 200 μg/mL of each functionalized nanosystem. Data represent 3 biological replicates with 2 technical replicates each with at least 1 million events, and are shown as mean ± SD (n = 6). Comparisons between

functionalized nanosystems and control MSN were done using ordinary one-way ANOVA with Dunnett's multiple comparisons test. Adjusted *p* values (from left to right): *M. marinum, M. fortuitum, M. smegmatis*: ****$p$ < 0.0001; *M. avium*: **$p$ = 0.0011 (TPP), ***$p$ = 0.0002 (AVA), ***$p$ = 0.0006 (AVA₂); *M. kansasii*: *$p$ = 0.0142 (TPP), **$p$ = 0.0095 (AVA), *$p$ = 0.0320 (AVA₂). **c** Representative Scan-ning Electron Microscopy (SEM) micrograph of *Mmar* cells after incubation with each nanosystem. The white arrows point out individual MSN. Scale bar: 2 μm. Two independent experiments were performed. Source data are provided as a Sour-ceData file.

Moreover, MSN-AVA₂-TPP showed again slightly less uptake, com-pared to MSN-TPP or MSN-AVA-TPP. This was further confirmed by the three-dimensional images, in which most of the MSN-AVA₂-TPP were located on the surface of the macrophage rather than within the cell, as opposed to MSN-TPP and MSN-AVA-TPP (Fig. 3c, Movies S1–S8). Those reasons, along with the better colloidal stability and higher affinity towards *M. marinum* in the presence of serum, made MSN-AVA-TPP the best candidate for further evaluations.

### MSN-AVA-TPP blocks *M. marinum* internalization in phagocytes
We next addressed whether MSN-AVA-TPP was able to block *Mmar* internalization in phagocytic cells, most specifically human phagocytic myeloid cells, as they are involved in the spread of the infection to the lymphatic system[17]. To do so, mScarlet-*Mmar* were pre-incubated with 50 and 200 μg/mL of FITC-labeled MSN or MSN-AVA-TPP for 15 min and then exposed for 30 min to human monocyte-derived macro-phages (MDM) and human monocyte-derived dendritic cells (MoDC) from three donors (Fig. 3d). Flow cytometry analysis showed that pre-incubating MSN-AVA-TPP and *Mmar* prior to the infection significantly impaired the uptake of *Mmar* inside MDM and MoDC compared to the untreated cells, while MSN alone did not have any significant effect (Fig. 3e). Overall, these results suggest that the physical contact between the nanoparticles and *Mmar* partially blocks the internaliza-tion of bacilli in MoDC and MDM, which consequently may participate in limiting the spread of the infection.

### In vitro activity of DOX-loaded MSN-AVA-TPP on *M. marinum*
The next step was to evaluate the effect of MSN-AVA-TPP on *Mmar* after loading them with an antibiotic. Since doxycycline (DOX) is a valuable agent in treating and managing skin, dental, respiratory, and urinary tract infections[31] including *Mmar* infection (Supplementary Table 2) (MIC = 1 μg/mL)[9], we loaded MSN-AVA-TPP with DOX, result-ing in MSN-AVA-TPP@DOX (Supplementary Fig. 5a). The nanoparticles were immersed in PBS to simulate the drug release in a physiologically relevant fluid. Overall, the nanoparticles released *ca.* 78 μg/mL (*ca.* 94% of the maximum released) in 9 h, observing an initial burst over the first 3 h (Supplementary Fig. 5b). The amount of DOX loaded in MSN-AVA-TPP was found to be *ca.* 28.16 ± 2.84 μg/mg of MSN-AVA-TPP. In addition, the data were fitted to a first-order kinetic model (Supple-mentary Fig. 5b, inset), shown in Equation 1: $Y = A(1 - e^{-kt})$, with $Y$ being the concentration of DOX released at a given time (t), $A$ the maximum amount of released DOX, and $k$, the release rate constant. The kinetics of DOX release appeared similar to the one reported for other antibiotic-loaded MSNs[32]. Moreover, the antimycobacterial effect of MSN-AVA-TPP@DOX was measured on *Mmar* planktonic cultures and biofilms. The minimal inhibitory concentration (MIC) and minimal bactericidal concentration (MBC) values corresponded to 125 and 250 μg/mL, respectively while the minimal biofilm inhibitory concentration (MBIC) and minimal biofilm eradication concentration (MBEC) values were 250 and 500 μg/mL, respectively (Supplementary Table 3). To demonstrate that our nanosystem can be loaded with

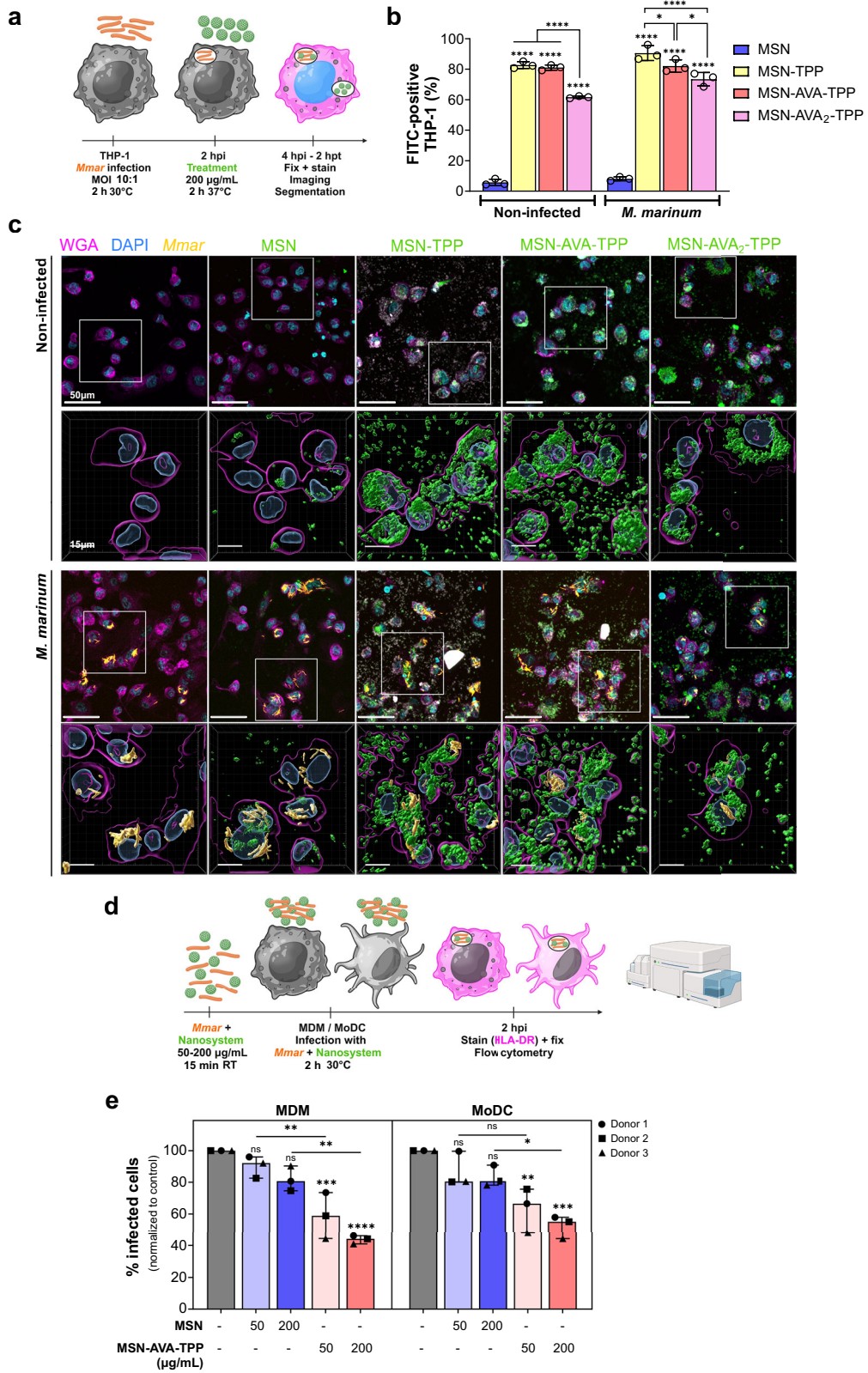

**a**

THP-1
*Mmar* infection
MOI 10:1
2 h 30°C

2 hpi
Treatment
200 µg/mL
2 h 37°C

4 hpi - 2 hpt
Fix + stain
Imaging
Segmentation

**b**

MSN
MSN-TPP
MSN-AVA-TPP
MSN-AVA$_2$-TPP

FITC-positive THP-1 (%)

Non-infected    *M. marinum*

**c**

WGA   DAPI   *Mmar*    MSN    MSN-TPP    MSN-AVA-TPP    MSN-AVA$_2$-TPP

Non-infected

50µm

15µm

*M. marinum*

**d**

*Mmar* +
Nanosystem
50-200 µg/mL
15 min RT

MDM / MoDC
Infection with
*Mmar* + Nanosystem
2 h 30°C

2 hpi
Stain (HLA-DR) + fix
Flow cytometry

**e**

Donor 1
Donor 2
Donor 3

MDM    MoDC

% infected cells
(normalized to control)

| MSN | - | 50 | 200 | - | - | | - | 50 | 200 | - | - |
| MSN-AVA-TPP (µg/mL) | - | - | - | 50 | 200 | | - | - | - | 50 | 200 |

other clinically relevant anti-mycobacterials, MSN-AVA-TPP nanoparticles were loaded with clarithromycin (CLR) (Supplementary Methods), one of the most important antibiotics in the treatment of non-tuberculous mycobacterial infections[33]. The kinetic of CLR release from CLR-loaded MSN-AVA-TPP nanoparticles was also validated (Supplementary Fig. 5c) and the MIC, MBC, MBIC and MBEC were

estimated (Supplementary Table 3), confirming the compatibility of our nanosystem with other antibiotics.

**MSN-AVA-TPP@DOX reduces the bacterial viability in biofilms**
The previously determined MBIC (250 µg/mL) and MBEC (500 µg/mL) were used to treat *Mmar* biofilms grown in granuloma-like medium

**Fig. 3 | Nanosystem affinity towards macrophages and effect on bacterial uptake. a** Schematic representation of the protocol used for the affinity assay. **b** Percentages of FITC-positive non-infected and *Mmar*-infected THP-1 cells after treatment with 200 μg/mL of each nanosystem for 2 h. The experiment was performed in 3 biological replicates and 3 technical replicates each. Data are shown as mean ± SD (n = 3). Comparisons were done using two-way ANOVA with Tukey's multiple comparisons test. Asterisks above each bar correspond to comparison to MSN within the condition. Adjusted *p* values (from left to right): Non-infected: ****$p < 0.0001$; *M. marinum:* ****$p < 0.0001$, *$p = 0.0171$ (TPP *vs* AVA-TPP), *$p = 0.0152$ (AVA-TPP *vs* AVA$_2$-TPP). **c** Representative images of non-infected and *Mmar*-infected THP-1 cells after treatment with 200 μg/mL of each nanosystem. First and third line correspond to xy orthogonal projection of z-stack confocal images, with a more detailed 3D reconstruction (second and fourth line) of the white squares. Three independent experiments were performed. **d** Schematic representation of the protocol used for the internalization assay. **e** Percentage of infected monocyte-derivate macrophages (MDM) (left panel) and monocyte-derivate dendritic cells (MoDC) (right panel) measured by flow cytometry. Both cell types were infected with mScarlet-*Mmar* pre-incubated with control (gray), 50 or 200 μg/mL of MSN (blue), or MSN-AVA-TPP (orange). The experiment was performed with 3 donors, and 3 technical replicates for each donor. Each dot corresponds to the mean of 3 technical replicates for one donor, normalized to the control. Data are shown as median and interquartile range (n = 3). Comparisons were done using ordinary one-way ANOVA with Šídák's multiple comparisons test. Asterisks above each bar correspond to comparison to non-treated condition (gray). Adjusted *p* values (from left to right): MDM: ns $p = 0.7783$ (MSN 50), ns $p = 0.1575$ (MSN 200), ***$p = 0.0008$, ****$p < 0.0001$, **$p = 0.0065$ (MSN 50 *vs* AVA-TPP 50), **$p = 0.0015$ (MSN 200 *vs* AVA-TPP 200); MoDC: ns $p = 0.5958$ (MSN 50), ns $p = 0.3277$ (MSN 200), **$p = 0.0049$, ***$p = 0.0006$, ns $p = 0.0814$ (MSN 50 *vs* AVA-TPP 50), *$p = 0.016$ (MSN 200 *vs* AVA-TPP 200). Created in BioRender. Kremer, L. (2025) https://BioRender.com/1qn1m1o. Source data are provided as a SourceData file.

(Fig. 4a), which allows the formation of a substratum for drug-tolerant mycobacterial biofilms[34]. Free DOX was also included at three different doses: 1.2 μg/mL (clinical concentration of DOX reaching the skin after oral administration of 1.125 ± 0.075 mg of DOX[35]), 6 μg/mL, and 12 μg/mL (theoretical concentrations of DOX released from 250 and 500 μg/mL of MSN-AVA-TPP@DOX, respectively). While treating the biofilm with 1.2 μg/mL of free DOX did not have a statistically significant effect compared to the control, treating the biofilm with 6 and 12 μg/mL of DOX reduced the bacterial burden by 55% and 69% respectively (Fig. 4b). Importantly, treating the biofilm with 250 or 500 μg/mL of MSN-AVA-TPP@DOX boosted the mycobactericidal effect (67% and 90%, respectively), suggesting that our nanosystem outperforms the free drug (Fig. 4b). We also looked for the effect of the nanosystem on cords, which result from the bacterial aggregation in a definite order and are considered as a major virulence factor in mycobacteria[36]. Moreover, cords have been shown to be the dominant structural phenotype within the *Mmar* biofilm[37]. By imaging the wells containing the *Mmar* biofilm, we observed smaller bacterial aggregates and cords when the biofilm was treated with 500 μg/mL of MSN-AVA-TPP@DOX, compared to the non-treated condition (Fig. 4c). Furthermore, confocal microscopy imaging on cords within the biofilm confirmed the attachment of the nanosystem to the bacteria within the biofilm (Fig. 4d), validating the efficiency of the antibiotic-loaded nanosystem against *Mmar* biofilms.

## MSN-AVA-TPP@DOX reduces *M. marinum* uptake in phagocytes
We then addressed whether MSN-AVA-TPP@DOX was able to block *Mmar* internalization in phagocytic cells as shown for MSN-AVA-TPP (Fig. 3d, e). To do so, mScarlet-*Mmar* were pre-incubated with 50 and 200 μg/mL of FITC-labeled MSN-AVA-TPP or MSN-AVA-TPP@DOX for 15 min and then exposed for 30 min to MDM and MoDC from three donors (Fig. 5a). Flow cytometry analysis showed that just like MSN-AVA-TPP, MSN-AVA-TPP@DOX reduced *Mmar* internalization (Fig. 5b), indicating that the physical contact between the nanosystem and *Mmar* was unaffected by the DOX loading.

## MSN-AVA-TPP@DOX impairs intracellular growth of *M. marinum*
The DOX-loaded nanoparticles were first incubated with THP-1 cells to assess toxicity. Three relevant doses were employed, namely MIC/MBIC (125 μg/mL), MBC (250 μg/mL) and MBEC (500 μg/mL). Overall, all doses showed comparable toxicity to the free drug (Supplementary Fig. 6), suggesting that the mild observed toxicity would stem from the drug itself, rather than from the nanomaterial. These therapeutically relevant doses were then employed to evaluate the effect of MSN-AVA-TPP@DOX treatment on *Mmar*-infected THP-1 cells (Fig. 5c). CFU counts after 24 h of treatment revealed that free DOX only produced statistically significant decrease of the intracellular bacterial burden at 12 μg/mL, while all concentrations of MSN-AVA-TPP@DOX did it in a dose-dependent manner (by 29%, 62% and 89%, respectively), showing that our system was 2.5 times more efficient than the free drug (Fig. 5d). The antimycobacterial features of MSN-AVA-TPP@DOX were further validated using confocal microscopy (Fig. 5e). As expected, MSN-AVA-TPP@DOX colocalized with intracellular *Mmar* (Fig. 5e, Movie S9–12). Moreover, while treatment with 125 μg/mL did not seem to have any effect on the intracellular bacilli (Movie S10), treatment with 500 μg/mL resulted in a more spherical mScarlet fluorescence loci (Fig. 6e, Movie S12), suggesting bacterial degradation. This confirmed that the nanosystem can be internalized by macrophages, reach intracellular compartments containing *Mmar* and, most importantly, largely outperforms the antimycobacterial effect of the free drug.

## MSN-AVA-TPP@DOX enhances survival of infected zebrafish
Zebrafish embryos are widely recognized for studying host-pathogen interactions[38] and provided major findings in *Mycobacterium* infection, on both mechanistic molecular studies related to host-pathogen interactions[39] and also in drug discovery research[40]. Therefore, FITC-labeled MSN-AVA-TPP@DOX was further validated in vivo using zebrafish embryos infected with *Mmar*, a natural fish pathogen. Initially, 2-day post-fertilization (dpf) embryos were injected in the caudal vein with either 10 ng or 20 ng of MSN-AVA-TPP or MSN-AVA-TPP@DOX or free DOX (500 pg corresponding to the theoretical amount of DOX released from 20 ng of MSN-AVA-TPP@DOX). Embryos were monitored for 6 days, showing no significant toxicity signs and validating the treatment doses (Supplementary Fig. 7a). A zebrafish transgenic line expressing *mcherry* in macrophages[41] was employed to assess nanoparticle location after injection, observing that they were rapidly engulfed by macrophages at 2 h post-treatment (hpt) (Fig. 6a, left panel). Almost all nanoparticles remained intracellularly after 24 h (Fig. 6a, right panel). We then established the mScarlet-*Mmar* infection by injecting around 80 CFU in the caudal vein of 30 hpf embryos. At 24 hpi, mycobacteria and cell aggregates (a signature of early granuloma formation) were observed (Fig. 6b). At that time point, the different treatments were administered, and the embryos were monitored for survival, CFU counts and imaging (Fig. 6c). Overall, untreated *Mmar*-infected embryos died after 8 days (Fig. 6d) upon developing an acute infection with increasing bacterial burden (Fig. 6e). While treatment with free DOX failed to affect the bacterial burden after 24 h, treatment with 10 ng or 20 ng of MSN-AVA-TPP@DOX reduced the burden by 81% and 89% respectively (Fig. 6e). Consequently, an increased survival rate of infected embryos of 25% (10 ng MSN-AVA-TPP@DOX) and 46% (20 ng MSN-AVA-TPP@DOX) was observed, suggesting a dose-dependent effect (Fig. 6d). No effect on survival was observed in embryos treated with free DOX or MSN-AVA-TPP (Fig. 6d). Imaging of infected embryos treated with MSN-AVA-TPP@DOX demonstrated the nanoparticle colocalization with bacteria (Fig. 6f, Movie S13), confirming the affinity of this nanosystem towards *Mmar* in vivo. Of note, treatment with the empty carrier also resulted in a slight decrease of

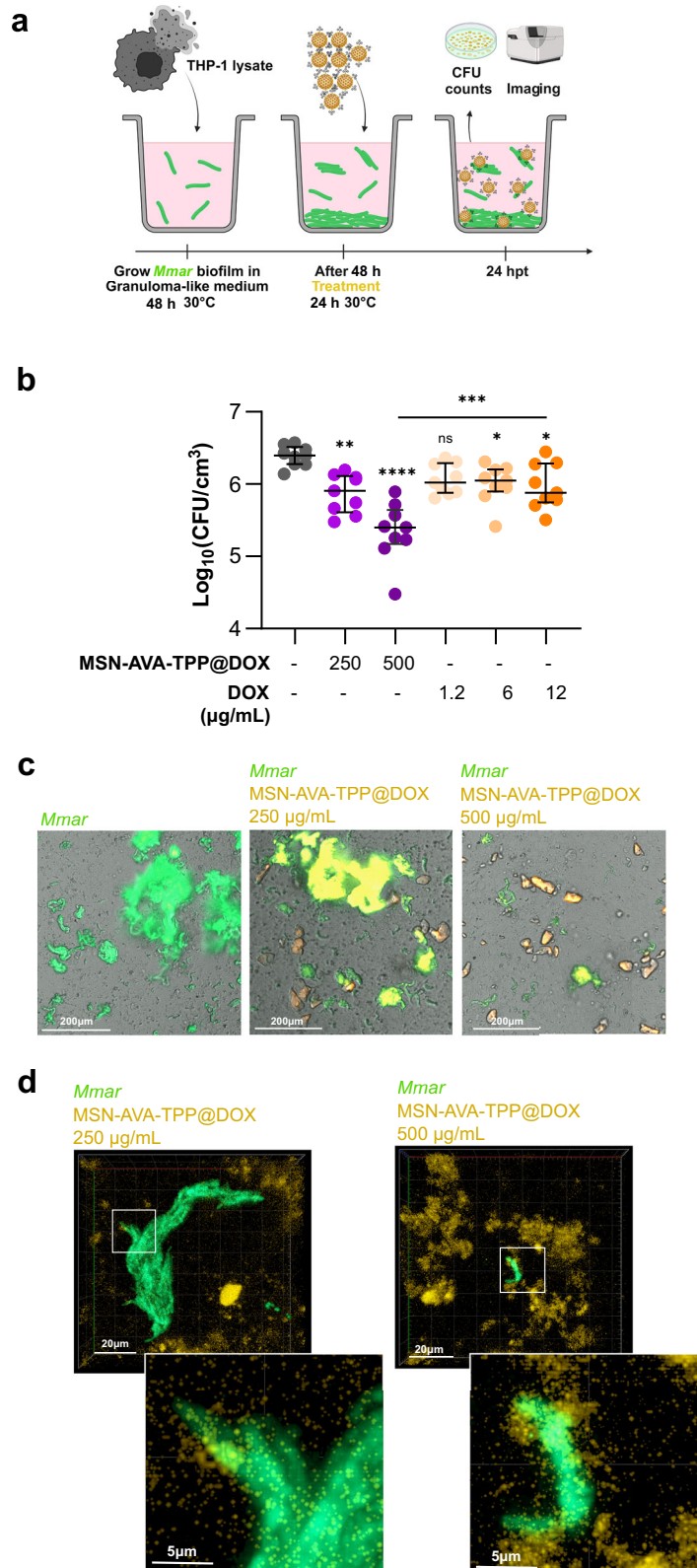

bacterial burden for 10 ng (20%) and 20 ng (39%) (Fig. 6e). This would agree with our previous findings that showed reduced bacterial internalization in cells when bacteria were pre-incubated with MSN-AVA-TPP (Fig. 3e).

Interestingly, CFU counts at 3 dpt revealed an increase in bacterial burden for all conditions (Fig. 6e). This suggests that while 20 ng MSN-AVA-TPP@DOX eliminated almost 90% of the bacteria after 24 h, the remaining bacilli could replicate and induce acute infection, resulting in the delayed embryo mortality. Increasing the dose of treatment could have solved this problem, nonetheless, we had already reached the maximum injection volume in the bloodstream and the maximum concentration of nanoparticles in 2% PVP without the risk of clogging the needle. Therefore, we decided to increase the frequency of injections. We first determined the clearance kinetics of MSN-AVA-

**Fig. 4 | Effect of MSN-AVA-TPP@DOX on *M. marinum* biofilms. a** Schematic representation of the protocol used. **b** Quantity of bacteria per granuloma-like medium volume represented as CFU/cm$^3$ (colony forming units), after treatment of the biofilm with MSN-AVA-TPP@DOX (250 and 500 µg/mL) or free DOX (1.2, 6, and 12 µg/mL). Four biological replicates were performed with at least 2 technical replicates each. Data are shown as individual counts (technical replicates) with median and interquartile range (n = 9). Comparisons were done using ordinary one-way ANOVA with Šídák's multiple comparisons test. Asterisks above each bar correspond to comparison to non-treated condition (gray). Adjusted *p* values (from left to right): \*\**p* = 0.0021, \*\*\*\**p* < 0.0001, ns *p* = 0.1328, \**p* = 0.0409 (DOX 6),

\**p* = 0.0202 (DOX 12), \*\*\**p* = 0.0001. **c** Representative images of the macro and microcolonies within the biofilm after no treatment (left panel), and treatment with 250 and 500 µg/mL of MSN-AVA-TPP@DOX (middle and right panel, respectively). Three independent experiments were performed. **d** Representative confocal images of microcolonies at different concentrations of MSN-AVA-TPP@DOX. First line corresponds to xy orthogonal projection of z-stack confocal images, with zoomed images (second line) of the white squares. Three independent experiments were performed. Created in BioRender. Kremer, L. (2025) https://BioRender.com/y4eccvw. Source data are provided as a SourceData file.

TPP@DOX in embryos by measuring the evolution of the FITC-labeled nanoparticles in terms of fluorescent pixel counts (FPC). After 48 h of treatment, we detected 69% less FITC signal *vs* 2 h and, after 72 h, this decrease reached 83% (Fig. 6g,h), suggesting a progressive nanoparticle clearance. We then tested different injection patterns to see if we could further increase the survival rate by continuously providing nanoparticles. Thus, infected embryos were administered one dose (at 1 dpi), two doses (at 1 and 3 dpi) or three doses (at 1, 3 and 5 dpi) of treatment. Increasing the frequency did not induce high toxicity signs (Supplementary Fig. 7b). Strikingly, this boosted even more the survival of infected embryos from 39% after one dose to 63% with the second dose and 89% with the third dose. This result resembled that of non-infected embryos that received the same treatment. Hence, the survival was directly proportional to the number of received doses. Imaging of embryos clearly showed numerous bacilli remaining in the infected zone and residues of MSN-AVA-TPP@DOX around these bacteria at 48 h after the first dose (Fig. 6j, left panel and Movie S14), compared to the massive reduction of the bacterial burden 48 h after the third one (Fig. 6j, right panel and Movie S15).

To further assess the applicability of our finding across other mycobacterial species, we tested our nanosystem on *M. fortuitum* (*Mfort*), after confirming the susceptibility of this strain to DOX (Supplementary Table 2). When treated with MSN-AVA-TPP@DOX, we found a MIC of 15.6 µg/mL, MBC of 62.5 µg/mL, MBIC of 31.2 µg/mL and MBEC of 250 µg/mL (Supplementary Table 4), confirming the activity of the system in vitro on *Mfort* planktonic growth and biofilms. Embryos were then infected with mScarlet-*Mfort*, according to a previous protocol[42]. Infected embryos were treated with either one dose of MSN-AVA-TPP@DOX at 1 dpi or three doses (1dpi, 3dpi, and 5dpi). Treated embryos responded to the treatment with enhanced embryo survival similar to non-infected controls (Supplementary Fig. S8). Collectively, this validates the antimycobacterial efficacy of MSN-AVA-TPP@DOX in vivo, highlighting the superior efficiency of this system over the free drug to strongly reduce mycobacterial infection and protect the embryos from death.

## Discussion

Current treatment options for *Mmar* infections imply long-term administration of antibiotics, thereby increasing the chances of superinfections and antibiotic resistance[10]. Here, we aimed to design a polyvalent nanomedicine capable of addressing *Mmar* infections in a comparatively shorter period. We engineered the surface of MSNs with a lipophilic cation able to: target the mycobacterial surface and block planktonic growth and biofilm formation (Fig. 7a(1)), enhance nanoparticles uptake by infected macrophages and efficiently eliminate the bacterial burden of infected macrophages (Fig. 7a(2)) and prevent further spreading by inhibiting bacterial uptake by human phagocytic myeloid cells (Fig. 7a(3)). Our nanosystem was also highly effective in vivo in a *Mmar*-zebrafish model of infection, resulting in a high rate of embryo survival (Fig. 7b).

The rationale behind the design of the different nanomaterials is two-sided: (1) positively-charged, TPP-modified molecules had previously been shown to target mycobacterial membranes, and (2) increasing the hydrophobicity of a drug conjugate may enhance its

interaction with bacteria[43]. Based on these premises, while we expected MSN-AVA$_2$-TPP to be the best candidate, flow cytometry (Fig. 2) and confocal microscopy (Fig. 3a-c) underscored MSN-AVA-TPP as the top candidate. This might be explained based on the surface chemistry (Fig. 1). TPP grafting efficacy on MSN-TPP outperformed its counterparts, attending to the higher percentage of weight loss despite not presenting AVA units. Even though MSN-AVA-TPP was shown to be the best candidate, the targeting capabilities of MSN-TPP and MSN-AVA-TPP were rather similar, suggesting that there might be a mixed contribution of charge and hydrophobicity on their capability to adhere to the mycobacterial surface. In addition to targeting isolated *Mmar*, MSN-AVA-TPP colocalize also with bacteria within infected macrophages (Fig. 3c) implying that, following internalization, the MSN-AVA-TPP particles have access to the *Mmar*-containing phagosome compartment and accumulate around the negatively-charged bacterial surface, thus endowing MSN-AVA-TPP with intracellular selectivity. Of note, the affinity towards the bacterial membrane was dictated by its specific composition, regardless of the nanoparticle type (Fig. 2a-b). This opens the door for finely tuning the surface functionalization to target the multiplicity of existing mycobacteria.

While it is generally accepted that *Mmar* is susceptible to rifampicin, ethambutol, clarithromycin, co-trimoxazole, DOX and minocycline[44], the use of DOX remains controversial[45,46]. Monotherapy based on DOX may be sufficient for early-stage, superficial limited infections of the skin but not for complicated cases where the infection has spread to deeper tissues[45]. Although the median duration of *Mmar* treatment with DOX is 91 days[47], a recent retrospective study showed that many patients had to interrupt DOX treatment owing to non-efficacy, relapse or side-effects in most of cases[46]. In this regard, we showed that 3 mg of DOX-loaded MSN-AVA-TPP released *ca.* 78 µg/mL in 9 h, meaning that 1 mg of nanoparticles releases 28 µg/mL of DOX. However, the clinically reachable DOX dose that can be achieved in the skin is 1.2 µg/mL after oral administration of 1 mg of free drug[35]. Thus, our nanoencapsulation approach leads to a dose increase of *ca.* 23-fold in a localized manner, which may shorten duration of the treatment and diminish the DOX-related side-effects in healthy organs and tissues.

In the granuloma, mycobacteria can live intracellularly[48,49] and extracellularly[50]. *Mmar* can survive inside macrophages, escape into the cytosol, and promote the direct cell-to-cell spread[51]. Here, we show that MSN-AVA-TPP@DOX significantly reduces the viability of intracellular mycobacteria living inside macrophages while producing minor toxicity to these cells (Fig. 5d, Supplementary Fig. 6). These results are in line with other nanosystems, e.g., polyanhydride nanoparticles loaded with rifampicin, isoniazid, pyrazinamide, and ethambutol against intracellular *Mmar* in macrophages[52], nanocarriers loaded with rifampicin and curcumin[53], colloidal Ag:ZnO mixture nanoparticles[54] or and CpG oligodeoxynucleotides-loaded porphyrin-based Zr-Metal-organic frameworks PCN-224 encapsulated with phosphatidylserine to treat intracellular *M. tuberculosis* in macrophages using phototherapy[55].

DCs are a bridge between the innate and acquired immunity[17]. In a granuloma or any inflammatory environment, MoDC differentiate in situ from monocytes[56]. Once DCs detect and phagocytose *Mmar*,

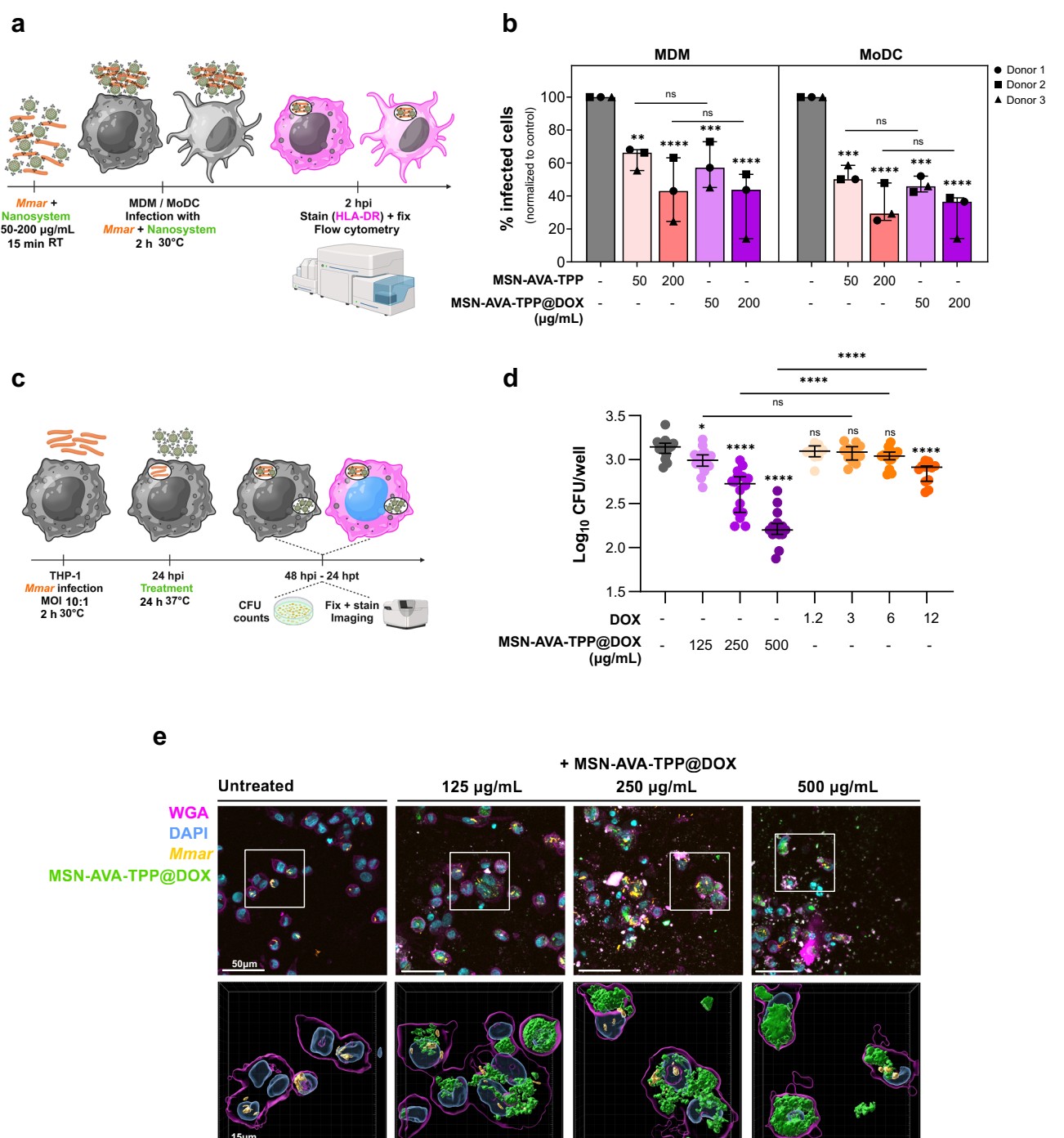

**Fig. 5 | Effect of MSN-AVA-TPP@DOX on *M. marinum* uptake and intracellular replication. a** Schematic representation of the protocol used for the internalization assay. **b** Percentage of infected monocyte-derivate macrophages (MDM) (left panel) and monocyte-derivate dendritic cells (MoDC) (right panel) measured by flow cytometry. Both cell types were infected with mScarlet-*Mmar* pre-incubated with control (gray), 50 or 200 µg/mL of MSN-AVA-TPP (orange), or MSN-AVA-TPP@DOX (purple). The experiment was performed with 3 donors, and 3 technical replicates for each donor. Each dot corresponds to the mean of 3 technical replicates for one donor, normalized to the control. Data are shown as median and interquartile range (n = 3). Comparisons were done using ordinary one-way ANOVA with Šídák's multiple comparisons test. Asterisks above each bar correspond to comparison to non-treated condition (gray). Adjusted *p* values (from left to right): MDM: **\*\*p* = 0.0026, \*\*\*\**p* < 0.0001, \*\*\**p* = 0.0007, ns *p* = 0.9995 (AVA-TPP 50 *vs* AVA-TPP@DOX 50), ns *p* = 0.9957 ((AVA-TPP 200 *vs* AVA-TPP@DOX 200); MoDC: \*\*\**p* = 0.0005 (AVA-TPP 50), \*\*\*\**p* < 0.0001, \*\*\**p* = 0.0002 (AVA-TPP@DOX 50), ns *p* = 0.9844 (AVA-TPP 50 *vs* AVA-TPP@DOX 50), ns *p* = 0.9987 (AVA-TPP 200 *vs* AVA-TPP@DOX 200). **c** Schematic

representation of the protocol used to quantify intracellular bacteria. **d** Quantity of bacteria per well of infected THP-1 cells, treated with different concentrations of MSN-AVA-TPP@DOX (125, 250, and 500 µg/mL) or free doxycycline (1.2, 3, 6, and 12 µg/mL). Three biological replicates were performed with at least 3 technical replicates each. Each dot corresponds to CFUs from one technical replicate. Data are represented as dot plots with median and interquartile range (n = 16 MSN-AVA-TPP@DOX; n = 12 DOX). Comparisons were done using ordinary one-way ANOVA with Šídák's multiple comparisons test. Asterisks above each data set correspond to comparisons to non-treated condition (gray). Adjusted *p* values (from left to right): *\*p* = 0.0416, \*\*\*\**p* < 0.0001, ns *p* = 0.9433 (DOX 1.2), ns *p* = 0.8660 (DOX 3), ns *p* = 0.2435 (DOX 6), ns *p* = 0.3928 (AVA-TPP@DOX 125 *vs* DOX 1.2). **e** Representative confocal images of *Mmar*-infected THP-1 cells treated with 125, 250, and 500 µg/mL of MSN-AVA-TPP@DOX. Left panels correspond to xy orthogonal projections of the z-stack images. Right panels correspond to zoomed 3D representation of the white squares. Three independent experiments were performed. Created in BioRender. Kremer, L. (2025) https://BioRender.com/zsckfm1. Source data are provided as a SourceData file.

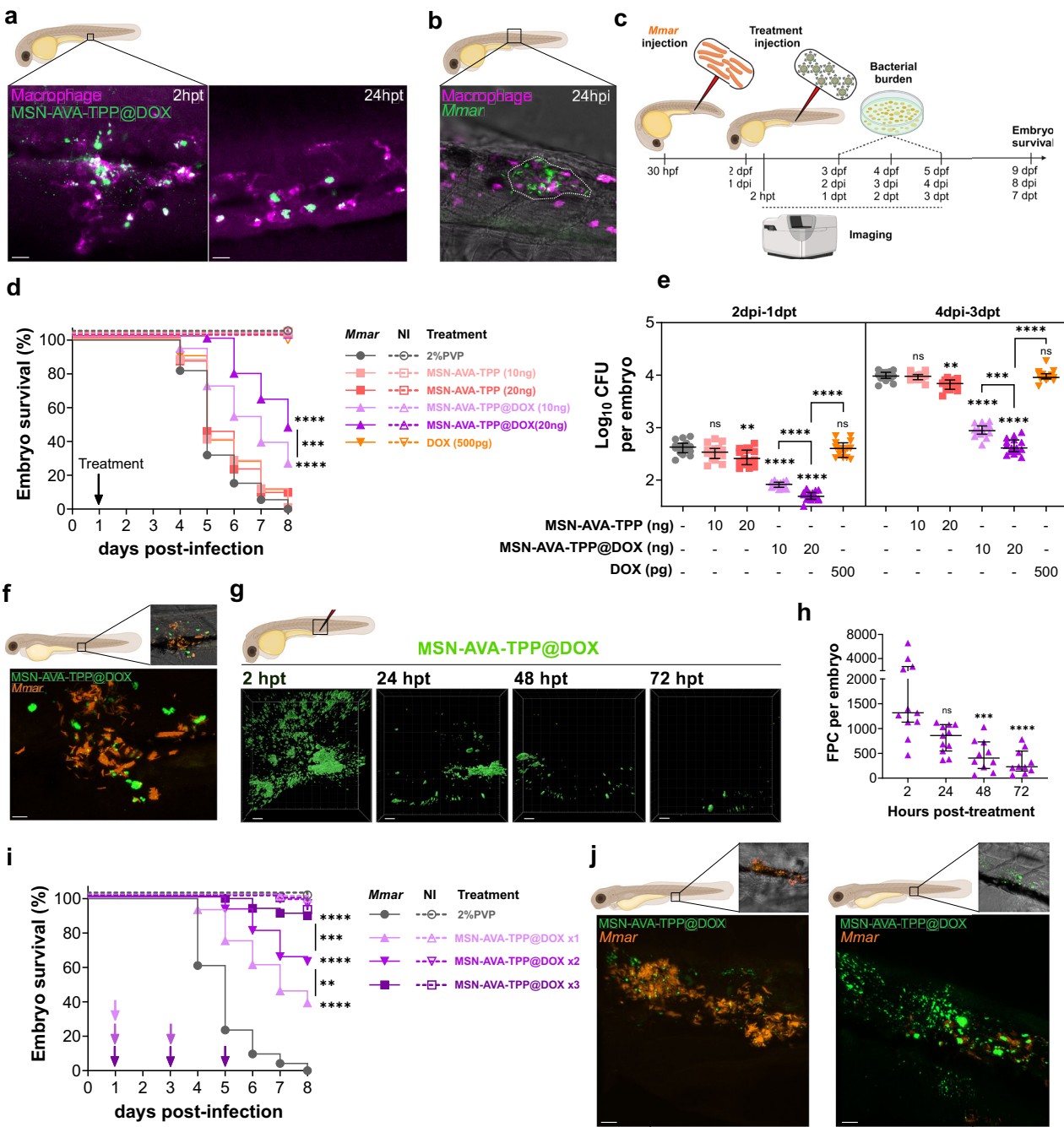

Fig. 6 | **Effect of MSN-AVA-TPP@DOX on infected zebrafish embryos with _M. marinum_. a** Localization of MSN-AVA-TPP@DOX after injection in embryos, 2 h post-treatment (hpt) and 24 hpt. Scale bar: 15 μm. **b** Representative image of an embryo with a pre-granuloma at 24 h post-injection (hpi). Scale bar: 20 μm. **c** Schematic representation of the protocol used. **d** Survival curve of infected embryos (full lines) with the different treatments injected at 24 hpi. Asterisks next to lines correspond to comparison with respective control (Dashed lines). _p_ values: ****_p_ < 0.0001, *** _p_ = 0.0005. **e** Bacterial burden from individual infected embryos with different treatments, at 2 dpi and 4 dpi. Median and interquartile range are shown (n = 15). Comparisons were done using ordinary one-way ANOVA with Šídák's multiple comparisons test. Asterisks above each data set correspond to comparison to non-treated condition. Adjusted _p_ values (from left to right): 2 dpi: ns _p_ > 0.9999, **_p_ = 0.0014, ****_p_ < 0.0001; 4 dpi: ns _p_ = 0.3739, **_p_ = 0.0020, ****_p_ < 0.0001, ns _p_ = 0.9990, ***_p_ = 0.0001. **f** Localization of MSN-AVA-TPP@DOX within an infection site at 24 hpt. Scale bar: 5 μm. **g** Kinetics of MSN-AVA-TPP@DOX after injection in embryos from 2 to 72 hpt. Images show 3D representation of

z-stack confocal images. Scale bar: 20 μm. **h** Quantification of MSN-AVA-TPP@DOX in embryos by fluorescent pixel counts (FPC). Median is shown with interquartile range (n = 12). Comparisons were done using Kruskal-Wallis with Dunn's multiple comparisons test. Asterisks above each data set correspond to comparison with 2 hpt. Adjusted _p_ values: ns _p_ = 0.1517, ***_p_ = 0.0005, ****_p_ < 0.0001. **i** Survival curve of infected embryos with either one, two or three doses of MSN-AVA-TPP@DOX (20 ng). Arrows indicate the time of treatment. Asterisks next to lines correspond to comparison with respective control. _p_ values: **_p_ = 0.0020, ***_p_ = 0.0004, ****_p_ < 0.0001. **j** Representative images of infection foci in infected embryos treated with either one dose of MSN-AVA-TPP@DOX or 3 doses, after 48 h of the last treatment. Scale bar: 10 μm. **a·b·d·e·f·g·h·i·j** Three independent experiments were performed with (**d–i**) n = 24 per condition and per experiment, comparisons were done using a log-rank (Mantel-Cox) test (n = 72 per condition), (**e**) n = 5 per condition per timepoint per experiment, (**g**, **h**) n = 4 per condition and per experiment (n = 16 per condition). Created in BioRender. Kremer, L. (2025) https://BioRender.com/h2chy8w. Source data are provided as a SourceData file.

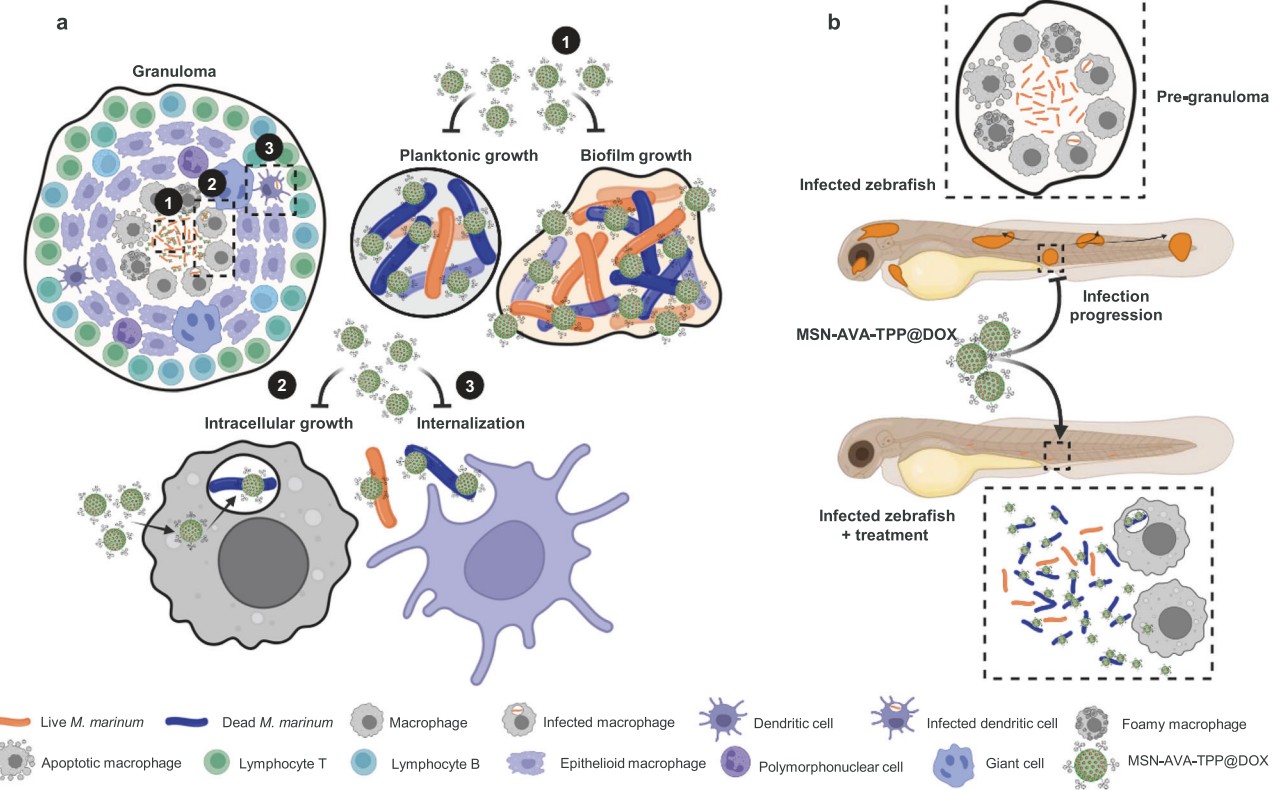

**Fig. 7 | Summary of our study and potential use of MSN-AVA-TPP loaded with doxycycline for the treatment of *M. marinum* infection. a** The in vitro studies show that MSN-AVA-TPP@DOX adheres onto mycobacteria in biofilms and plank-tonic state present in the caseum. The nanosystem reduces the viability of some mycobacteria within the biofilms (1) and is internalized in the *Mmar*-infected macrophages, reaching the intracellular mycobacteria, and kill them (2). These two abilities of the nanosystem would inhibit the *Mmar* infection progression from the granuloma to deeper tissue layers. In addition, mycobacteria which interact with the nanosystem would also not be phagocytosed by macrophages and dendritic cells (3), thus limiting the sporotricoid-like lymphatic progression associated with *Mmar* infection. **b** Schematic representation the infection course of a zebrafish embryos infected with *Mmar*. The infection can progress from a first pre-granuloma (dashed square) to other infection foci (embryo on top). Treatment with MSN-AVA-TPP@DOX blocks the infection progression by killing not only extracellular bacteria, but bacteria inside macrophages as well, limiting the spread of the infection and resulting in embryos survival (lower embryo). Created in BioRender. Kremer, L. (2025) https://BioRender.com/48jwn1f.

they undergo a maturation process and increase the expression of major histocompatibility complex class I and II molecules, costimulatory molecules, among others, to drive effective immunity[57]. Such phenotypic changes allow *Mmar*-infected DCs to migrate from the granuloma to the draining lymph nodes[17], promoting the typical sporotricoid-like spread related to *Mmar*[58]. Figure 3e and Fig. 5b points out that the interaction between MSN-AVA-TPP and *Mmar* reduces the percentage of infected MoDC in a dose-dependent manner. Hence, it can be inferred that MSN-AVA-TPP creates a physical barrier that prevents bacterial recognition by MoDC. Similar conclusions may originate from the dose-dependent reduced bacterial burden observed in zebrafish embryos treated with drug-free MSN-AVA-TPP (Fig. 6e). To the best of our knowledge, this is the first time that an engineered nanomaterial is shown to impair bacterial uptake by immune cells per se, which may contribute to reduce the sporotricoid-like spread of *Mmar* from the granuloma. Granulomas have lately been linked to biofilm infections[59]. Several mycobacteria, including *Mmar*, can form biofilms within in vitro and in vivo granulomas[60]. Recently, it was demonstrated that mycobacteria can develop biofilm in RPMI 1640 supplemented with the lysate of leukocytes[61], providing an in vitro model mimicking the environment of the granuloma caseum[34]. Even though the mycobacterial biofilms grown in this medium are highly intolerant to antibiotics, we proved that MSN-AVA-TPP@DOX attaches to the biofilm and reduces the viability of *Mmar* within the biofilm (Fig. 4).

Zebrafish is a natural host of several mycobacterial species and a surrogate model organism for *Mmar* infection research[39,62]. Several studies have used the zebrafish to investigate skin infections, e.g., *Yersinia ruckeri*[63], *Staphylococcus aureus*[64], *Aeromonas hydrophila*[65], and, most importantly *Mmar*-induced skin inflammation[66]. In addition, zebrafish has been employed for testing the toxicological and biodistribution profile of nanoparticles, including antimycobacterial nanoparticles[67]. *Mmar*-infected embryos rapidly develop an acute infection, which is usually fatal within a week[68]. Taking advantage of this model, we show here that treatment with MSN-AVA-TPP@DOX not only reduced the bacterial burden but also enhanced embryo survival in a dose-dependent manner. Furthermore, we demonstrated that increasing the dose frequency achieved almost full embryo survival without affecting the overall health of the animals (Fig. 6).

In summary, the screening of a series of engineered MSNs resulted into a nanocarrier displaying very high activity against *Mmar* infections, capable of efficiently targeting the mycobacterial surface, hampering subsequent bacterial uptake by macrophages and DCs. The antimycobacterial features of these nanocarriers have been validated using DOX-loaded nanoparticles, demonstrating bacterial elimination both in vitro and in vivo. Importantly, a three-dose treatment achieved almost complete survival of the infected zebrafish embryos. Together, this work establishes that engineering MSN with TPP derivatives represents a valuable approach to treat mycobacterial infections, with a marked emphasis on *Mmar* infections. Hence, our findings not only

support the use of these nanoparticles as a promising local treatment of isolated *Mmar* granulomas but also to a wider range of cutaneous bacterial infections.

# Methods

## Ethics statement
Zebrafish (*Danio rerio*) were handled in compliance with European Union guidelines for laboratory animal care, approved by the Direction Sanitaire et Vétérinaire de l'Hérault for the ZEFIX-CRBM zebrafish facility (Montpellier) under registration number C-34-172-39. Experimental protocols were approved by Le ministère de l'Enseignement Supérieur, de la Recherche et de l'Innovation under the reference APAFIS#24406-2020022815234677 V3.

## Synthesis of mesoporous silica nanoparticles (MSN)
The following reagents were purchased from Sigma-Aldrich: Tetraethyl orthosilicate (TEOS); Sodium hydroxide (NaOH); Ammonium Nitrate; Hexadecyltrimethylammonium bromide (CTAB); Rhodamine B isothiocyanate (RITC); Fluorescein isothiocyanate (FITC); 3-(Aminopropyl)triethoxysilane (APTES); (4-Carboxybutyl)triphenylphosphonium bromide (TPP); 5-(Fmoc-amino)valeric acid (AVA-Fmoc); N-(3-(dimethylamino)propyl)-N'ethylcarbodiimide hydrochloride (EDC); N-hydroxysuccinimide (NHS); Piperidine; ethanol; N,N-dimethylformamide (DMF). MSN were synthesized following a previously described method[69]. Briefly, CTAB (1 g, 2.74 mmol) was placed in a 1 L, round-bottom flask. Then, distilled water (480 mL) and NaOH (3.5 mL, 2 M) were added, and the mixture was heated to 80 °C under magnetic stirring. TEOS (5 mL, 22.39 mmol) was then added dropwise (0.25 mL/min) and the mixture was heated at 80 °C for further 2 h. The reaction was then cooled down in an ice bath, centrifuged and washed twice with a 1:1 mixture of ethanol/water (18,630 x g, 10 °C, 15 min). The CTAB template was removed by ionic exchange, using an ethanolic aqueous solution (95% in ethanol) of $NH_4NO_3$ (10 g/L). The previously obtained precipitate was redispersed in 350 mL of this solution, refluxed for 2 h, centrifuged, and washed once with a 1:1 mixture of ethanol/water (18,630 x g, 10 °C, 15 min) and the process was repeated twice. Finally, the surfactant-free nanoparticles were redispersed in absolute ethanol and kept in the dark until further functionalization. The nanoparticles were labeled for the different biological experiments. FITC (0.78 mg, 0.002 mmol) or RITC (1.07 mg, 0.002 mmol) were reacted with APTES (2.2 μL, 0.009 mmol) in ethanol (40 μL) for 2 h. The mixture was then carefully mixed with TEOS (5 mL, 22.39 mmol) and the nanoparticles were produced following the methodology described above.

## Synthesis of aminated MSN (MSN-NH₂)
The as-synthesized MSN (275 mg) were placed in a two-neck, round-bottom flask equipped with a reflux. The particles were dried at 100 °C under vacuum to remove water traces from the sample. Then, a N₂ atmosphere was introduced, and the powder was redispersed in anhydrous toluene (10 mg/mL). After that, APTES (100 μL, 0.43 mmol) in anhydrous toluene (1 mL) was added dropwise and the whole reaction mixture was heated at 80 °C overnight. The nanoparticles were collected by centrifugation and subsequently washed three times with ethanol (18,630 x g, 10 °C, 15 min). The final precipitate was dried in an oven at 70 °C overnight.

## Synthesis of TPP-functionalized MSN (MSN-TPP)
The as-synthesized MSN-NH₂ (25 mg) were placed in a vial and dispersed in distilled water (10 mg/mL). Separately, TPP (25 mg, 0.056 mmol), EDC (16.2 mg, 0.084 mmol) and NHS (12.95 mg, 0.112 mmol) were dissolved in the minimum amount of distilled water and stirred for 3 h. Then, the latter was added dropwise to the previously dispersed nanoparticles, and the mixture was stirred overnight at room temperature. The nanoparticles were centrifuged and washed 3 times with ethanol (18,630 x g, 10 °C, 15 min) and dried in an oven at 70 °C overnight.

## Synthesis of (AVA)ₙ-TPP-functionalized MSN (MSN-(AVA)ₙ-TPP)
All materials were produced sequentially, starting from MSN-NH₂ and progressively incorporating the desired AVA units. TPP was added once the corresponding number of AVA units was successfully grafted to the nanoparticles. For that purpose, MSN-NH₂ (60 mg) were placed in a vial and dispersed in DMF (10 mg/mL). Separately, AVA-Fmoc (50 mg, 0.147 mmol), EDC (42.36 mg, 0.221 mmol) and NHS (33.91 mg, 0.294 mmol) were dissolved in the minimum amount of DMF and stirred for 3 h. Then, the latter was added dropwise to the previously dispersed nanoparticles, and the mixture was stirred overnight at room temperature. The nanoparticles were centrifuged, washed with DMF twice (18,630 x g, 10 °C, 15 min) and subsequently resuspended in 1 mL of piperidine (20% in DMF) for Fmoc deprotection. The nanoparticles were stirred for 4 h, centrifuged and washed twice with DMF, yielding MSN-AVA-NH₂, subsequently split in two groups for either further AVA-Fmoc addition or TPP grafting. The AVA grafting protocol described in this section was repeated, yielding MSN-AVA₂-NH₂. Finally, each group (15 mg) was dispersed in distilled water. Separately, TPP (15 mg, 0.034 mmol), EDC (9.72 mg, 0.051 mmol) and NHS (7.77 mg, 0.068 mmol) were dissolved in the minimum amount of distilled water and stirred for 3 h. Then, the latter was added dropwise to the previously dispersed nanoparticles, and the mixture was stirred overnight at room temperature. Finally, the nanoparticles were centrifuged and washed three times with ethanol (18,630 x g, 10 °C, 15 min), and dried in an oven at 70 °C overnight, yielding MSN-AVA-TPP and MSN-AVA₂-TPP, respectively.

## Antibiotic loading and release
The different nanoparticles were loaded by shaking them in the presence of a saturated solution of the corresponding antibiotic. To produce DOX-loaded nanoparticles, 4–6 mg of MSN-AVA-TPP were dispersed in 1 mL of a 2.5 mg/mL solution of doxycycline monohydrated (Sigma Aldrich, USA) in methanol (Honeywell, USA). The solution was stirred at 186 ×g and 4 °C overnight. The antibiotic-loaded nanoparticles were centrifuged, and the supernatant was removed. The DOX-loaded nanoparticles were dried at room temperature for at least 5 h under a chemical hood. To determine the kinetic of drug release from the nanoparticles, 12 mg of the DOX-loaded MSN-AVA-TPP were suspended in 2 mL of Phosphate Buffered Saline (PBS, Sigma Aldrich, USA). Suspensions were placed into the lower chamber of a 12-well plate Transwell® (0.4 μm-diameter pore, Corning, USA) while 1 mL PBS was placed in the upper chamber. PBS was used because it is standard solution for evaluating antibiotic release from nanomaterials[69–71]. The final concentration of nanoparticles was kept at 3 mg/mL per well (n = 3) and the plate incubated at 37 °C. Periodically, 200 μL were collected from the upper chambers and replaced by 200 μL of fresh PBS. The DOX concentration was determined by measuring the absorbance at 347 nm[72] in a spectrophotometer multimode microplate reader (Tecan Spark 10 M; Tecan Group Ltd., Switzerland).

## Bacterial strains and growth conditions
Bacterial strains used in this study are listed in Supplementary Table 1. All strains were grown in Middlebrook 7H9 (BD, USA) supplemented with 10% oleic acid, albumin, dextrose, and catalase (OADC enrichment), 0.025% tyloxapol (Sigma Aldrich, USA) and the respective antibiotic. Wild-type *M. marinum* strain M[73] was transformed with either pMV306-$P_{left}$*mScarlet or pMV306-$P_{left}$*mWasabi[74], yielding mScarlet-*Mmar* and mWasabi-*Mmar*, respectively. *M. fortuitum* was transformed with pMV306-$P_{left}$*mScarlet yielding mScarlet-*Mfort*. Strains were kept at -80 °C until further use or grown in 7H9 supplemented with 10% OADC, 0.025% tyloxapol and 50 μg/mL

kanamycin when required or plated on Middlebrook 7H11 (Merck, USA) supplemented with 10% OADC with 50 µg/mL kanamycin when required, at 30 °C for at least 5 days for *Mmar*, and 37 °C for at least 48 h for *Mfort*.

### Antibiotic susceptibility testing

Antimicrobial susceptibility of both fluorescent strains was tested using broth microdilution following the European Committee on Antimicrobial Susceptibility Testing (EUCAST)[75]. For this purpose, 96-well RAPMYCOI Sensititre™ titration plates (Thermo Fisher Scientific, Massachusetts, USA) were used following the manufacturer's recommendations and using an incubation at 30 °C for 7 days.

### Mycobacteria-nanoparticle interactions

The different mycobacterial cultures (Supplementary Table 1) were washed with 0.9% NaCl (Physiodose, Laboratoire Gilbert, France) and resuspended at a concentration of $10^7$ colony-forming units (CFU)/mL in 0.9% NaCl supplemented with 0.025% tyloxapol with or without 10% human serum. FITC-labeled nanoparticles were incubated with the mycobacterial suspensions at 200 µg/mL at room temperature in an Eppendorf Thermomixer R (Eppendorf, Germany) for 5 min under agitation (186 ×*g*). Samples were then analysed by flow cytometry using a NovoCyte ACEA flow cytometer. The gating strategy is presented in Supplementary Fig. 1. Briefly, gates were drawn using SSC-A/FSC-A. FITC-labeled MSN (200 µg/mL), non-fluorescent mycobacteria or mycobacteria expressing mScarlet or tdTomato were included as controls. Analysis was done with the NovoExpress version 1.6.2. Experiments were performed by using three biological replicates and two technical replicates with at least 1 million events measured per technical replicate.

### Scanning electron microscopy (SEM)

The interaction between bacteria and nanoparticles was evaluated using SEM. Sample were resuspended in a 12-well Transwell® insert (0.4 µm-diameter pore) and placed on absorbing paper to remove the liquid phase. Mycobacteria were washed four times with 1 mL of sterile PBS, resuspended in 1 mL of PBS and centrifuged before fixation in 1 mL of 4% paraformaldehyde (PFA) in PBS at 4 °C overnight. The samples were washed in PBS and dehydrated using a gradient of increasing ethanol (30, 50, 70, 90, 100%) for 10 min and the dehydrated samples were resuspended in pure hexamethyldisilazane. Finally, the samples were sputter coated with a *ca*. 10 nm thick gold film and then examined under a Hitachi S4000 scanning electron microscope using a lens detector with an acceleration voltage of 10 kV at calibrated magnifications.

### Determination of MIC and MBC

Minimum inhibitory concentrations (MIC) were determined using a previously described broth microdilution method[76] with one modification. The MIC is the minimum concentration required to inhibit the bacterial visible growth. In brief, concentrations of MSN-AVA-TPP ranging from 500 µg/mL to 1.973 µg/mL with a two-fold dilution were added to Cation-adjusted Mueller-Hinton broth (Sigma Aldrich, USA) (CaMHB) to a final volume of 100 µL/well. One hundred microlitres of mScarlet-*Mmar* in CaMHB containing approximately $3 \times 10^6$ CFU/mL were added to a 96-well Clear Round Bottom TC-treated Cell Culture Microplate (Corning Inc., USA) followed by static incubation at 30 °C for at least 7 days. After incubation, MIC was determined visually the lowest concentration of nanoparticles where no bacterial growth was observed. Minimum bactericidal concentration (MBC) was determined using the flash microbiocide method[77]. The MBC is defined as the minimum concentration required to kill bacteria. Briefly, 20 µL of the corresponding well from the MIC 96-wells plate were mixed with 180 µL of Middlebrook 7H9 supplemented with OADC and tyloxapol in a new 96-well plate, which was then incubated statically at 30 °C for

5 days. After incubation, MBC was determined visually as the lowest concentration of nanoparticles where no bacterial growth was observed. The experiments were performed using three biological replicates.

### Determination of MBIC and MBEC

Minimal biofilm inhibitory concentrations (MBIC) and minimal biofilm eradication concentrations (MBEC) were determined using a previously described methodology[78]. The MBIC is the minimum concentration required to inhibit the visible growth of a bacterial biofilm. For the MBIC, a biofilm was formed by inoculating 100 µL of CaMHB containing $3 \times 10^6$ CFU/mL of bacteria on the bottom of the wells of a Nunc™ MicroWell™ 96-well, non-treated, flat-bottom plate (Thermo Fisher Scientific, United States) and the plate was statically incubated at 30 °C for 48 h prior to removal of the supernatant. Afterwards, each well was filled with 200 µL of CaMHB containing different concentrations of the corresponding nanoparticles, ranging from 500 µg/mL to 1.973 µg/mL with a two-fold dilution and the plate was statically incubated at 30 °C for 5 days. The MBIC was determined visually as the lowest concentration of nanoparticles where no planktonic bacterial growth from the biofilm was observed. For the MBEC, the bottom of each well was scrapped with a 100 µL tip to physically detach the biofilm from the bottom surface of each well. Then, 20 µL of each well were transferred to a new well containing 180 µL of Middlebrook 7H9 supplemented with OADC and tyloxapol and the plate was statically incubated at 30 °C for 5 days. The MBEC was determined visually as the lowest concentration of nanoparticles where no mycobacterial growth was observed. The experiments were performed using three biological replicates.

### Treatment of *Mmar* biofilms grown in granuloma-like medium

*Mmar* biofilms were grown using a modification of a previously described methodology[61]. One hundred microlitres of mWasabi-*Mma*r strain ($3 \times 10^7$ CFU/mL) were added to sterile 2 mL tubes in complete RPMI-1640 supplemented with 2% heat-inactivated fetal bovine serum (HI-FBS) and $7.5 \times 10^6$ cells per millilitre of THP-1 monocytes lysed by freezing at -80 °C. The tubes were incubated at 30 °C for two days. After incubation, DOX-loaded (MSN-AVA-TPP@DOX), rhodamine-B labeled nanoparticles resuspended in RPMI with HI-FBS were added to each biofilm at 250 and 500 µg/mL. Treatment with free DOX at 1.2, 6 and 12 µg/mL was also included. The tubes were incubated at 30 °C for 24 h. Then, 800 µL PBS with 0.025% tyloxapol were added to each tube, sonicated for 5 min with an ultrasonic bath (Bandelin Sonorex Super RK 255 H, Sigma Aldrich) until complete homogenization. The mycobacterial suspension was serially diluted on Middlebrook 7H11 supplemented with OADC and incubated at 30 °C for 5 days. Data are expressed as CFU/cm³. This experiment was performed with four biological replicates and two technical replicates at least (n = 9). Representative images of the biofilm were acquired performing the same experiment in µ-Slide 8-well chambers (ibidi, Gräfelfing, Germany) using a Cell-Discoverer 7 (Carl Zeiss SAS, France) with a Plan-Apochromat 5×/0.35 objective and 0.5× tubulens optovar in a Widefield mode, and more detailed 3D confocal images were acquired with a Plan-Apochromat 50×/1.2 water objective and 1× tubulense optovar, by taking z-stack images with a range between 10 and 15 µm (0.215 µm slices). Images were processed using Zen Blue 3.2 software (Zeiss).

### Cell culture

The human pro-monocytic cell line THP-1 was maintained in RPMI 1640 GlutaMAX™ supplemented with 10% fetal bovine serum (FBS) and 1% penicillin/streptomycin (Gibco™) and kept in a 5% $CO_2$ incubator at 37 °C. THP-1 cells were differentiated into macrophages following 72 h with 20 ng/mL of phorbol myristate acetate (PMA). Human monocytes from buffy coats were obtained according to the institutional

guidelines of the Ethical Committee of the CNRS and EFS. After Ficoll gradient on PBMC, monocytes were isolated using CD14 MicroBeads (Miltenyi Biotec). Usual purity was >95% CD14$^+$. Human monocyte-derived dendritic cells (MoDC) and monocyte-derived macrophages (MDM) were generated by incubating purified monocytes in IMDM-GlutaMAX™ supplemented with 10% fetal calf serum, 100 IU/mL penicillin, 100 µg/mL streptomycin, 10 mM Hepes, 100 µM non-essential amino acids, 1 mM sodium pyruvate, 25 µM 2-mercap-toethanol, and adding either GM-CSF (500 IU/mL) and IL-4 (500 IU/mL) for MoDC or M-CSF (50 ng/mL) for MDM. The obtained immature MoDC and MDM were harvested between days 5 and 7 and pheno-typed by flow cytometry before experimental use. The following antibodies were used to phenotype myeloid cells (MoDC and MDM) by flow cytometry: CD1a-FITC (#300103), CD14-PE (#301806), CD83-FITC (#305305) and HLA-DR-APC (#980406) were from Biolegend. The anti-CD80-PE (#TNB50-0809-T025) was from TONBO (Cytek) and the anti-CD209-APC (#130-124-257) was from Miltenyi Biotec. The following isotype antibodies mouse IgG1k-Isotype-FITC (#11-4714-42), mouse IgGk-Isotype PE (#12-4714-42) and mouse IgGk-Isotype APC (#17-4714-82) were from Thermofisher and used as staining control. Tissue cul-ture supplements were purchased from Sigma-Aldrich, while cytokines were from Miltenyi Biotec. For each experimental approach, three donors were used for 3 biological replicates.

### Nanoparticle affinity towards infected macrophages
THP-1 were seeded in PhenoPlate™ 96-well microplates with black walls and cyclic olefin bottoms (Revvity) ($3 \times 10^4$ cells/well) and differentiated as described above. After 72 h, cells were washed 3 times and infected with mScarlet-*Mmar* at a multiplicity of infection (MOI) of 10 in EM medium (120 mM NaCl, 7 mM KCl, 1.8 mM CaCl$_2$, 5 mM Glucose, 25 mM HEPES pH 7.3) for 2 h at 30 °C[79]. Cells were then washed 3 times with PBS and treated with 200 µg/mL of MSN, MSN-TPP, MSN-AVA-TPP, and MSN-(AVA)$_2$-TPP, for 2 h at 37 °C in EM medium supplemented with 10% FBS and 50 µg/mL amikacin. Cells were then washed 3 times with PBS, fixed with 4% PFA for 15 min at room temperature, and labeled with Wheat-Germ-Agglutinin (WGA), Alexa Fluor™ 647 conjugate (Invitro-gen) to stain the cell membranes, as well as DAPI for the cell nucleus. Cells were imaged using a Cell-Discoverer 7 with a Plan-Apochromat 5x/0.35 objective and 0.5x tubulens optovar, using Tiles to image the full well. Representative three-dimensional confocal images were then acquired using a Plan-Apochromat 20 × /0.7 objective and 1× tubulens optovar by taking z-stack images with a range between 10 and 15 µm. Images were segmented using Zen Blue 3.2 software (Zeiss) as followed: Cells were identified as a class of objects using WGA and DAPI seg-mentation, intracellular bacteria were identified as a subclass within the cells using the mScarlet segmentation, and intracellular nanoparticles were identified as subclass within the cells using the FITC segmentation. Around $3 \times 10^5$ cells were segmented per condition. The experiment included three biological replicates with three technical replicates (3 wells) per condition per experiment (n = 9). Surface representation was done using Imaris 10.1.1.

### *M. marinum* internalization assay in phagocytic cells
MoDC and MDM were seeded in U-bottom 96-well plates ($2 \times 10^4$ cells/well). Prior to infection, mScarlet-*Mmar* were incubated with 50 and 200 µg/mL of MSN or MSN-AVA-TPP or MSN-AVA-TPP@DOX for 15 min at room temperature. Cells were infected at a MOI of 100 for 30 min in EM medium at 30 °C and then fixed for 15 min with 4% PFA. Cells were then labeled with HLA-DR Alexa Fluor™ 647 conjugate for 1 h at room temperature and washed 3 times with PBS supplemented with 5 mM of EDTA. Samples were analysed by flow cytometry using a NovoCyte ACEA flow cytometer. Gates were drawn using SSC-A/FSC-A. Non-infected cells as well as cells infected with mScarlet-*Mmar* without pre-incubation with MSNs were used as negative and positive controls, respectively. Analysis was done with NovoExpress version 1.6.2.

Experiments were performed by using three biological replicates (three donors) and three technical replicates for both cell types.

### Macrophage cytotoxicity assay
THP-1 cells were differentiated into macrophages following incubation with 20 ng/mL PMA in 96-well plate ($2 \times 10^4$ cells/well) for 72 h. Then, a series of nanoparticle concentrations ranging from 1 mg/mL to 7.81 µg/mL were added to the cells and incubated for either 24 h or 72 h. At the desired time point, 10 % v/v resazurin was added to each well and left to incubate for 4–6 h at 37 °C with 5% CO$_2$. Absorbance was measured using a fluorescence plate reader (excitation 540 nm, emission 590 nm). Non-treated cells were used as a control. To evaluate the cytotoxicity of MSN-AVA-TPP loaded with doxycycline, THP-1 were seeded into 96-well plates ($3 \times 10^4$ cells/well) and differentiated as described above. After 72 h, cells were washed and treated with MSN-AVA-TPP@DOX at 500, 250, and 125 µg/mL, or free DOX at 1.2, 3, 6, and 12 µg/mL for 24 h. Cytotoxicity was measured using CellTiter 96® AQ$_{ueous}$ One Solution Cell Proliferation Assay (MTS) (Promega) fol-lowing the manufacturer's recommendations. Non-treated cells were used as a control.

### Macrophage infection and treatment with MSN-AVA-TPP@DOX
THP-1 were seeded in PhenoPlate™ 96-well microplates with black walls and cyclic olefin bottoms (Revvity) ($3 \times 10^4$ cells/well) and differ-entiated as described above. After 72 h, cells were washed 3 times and infected with mScarlet-*Mmar* at a MOI of 10 in EM medium (NaCl 120 mM, KCl 7 mM, CaCl$_2$ 1.8 mM, Glucose 5 mM and HEPES 25 mM, pH 7.3) for 2 h at 30 °C. Cells were then washed 3 times with PBS and kept in EM supplemented with 10% FBS and 50 µg/mL of amikacin at 30 °C in the presence of 5% CO$_2$. After 24 h, cells were treated with MSN-AVA-TPP@DOX at 125, 250 and 500 µg/mL and maintained at 30 °C in the presence of 5% CO$_2$. DOX was used as a control at 1.2 µg/mL repre-senting 1-fold the therapeutically reachable concentration of DOX[35], 3, 6, and 12 µg/mL representing the theoretical amount of DOX released from 125, 250, and 500 µg/mL of MSN-AVA-TPP@DOX, respectively, according to our release experiment. 24 h after the treatment, cells were washed three times with PBS and lysed with 0.1% Triton-X-100 for 10 min to recover intracellular mycobacteria. Lysates were 10-fold diluted in PBS supplemented with 0.025% tyloxapol and plated on Middlebrook 7H11 + OADC and kanamycin 50 µg/mL. Plates were incubated at 30 °C and colonies were counted after 7 days. Data are expressed as Log$_{10}$ CFU/mL. The experiments were performed by using 3 biological replicates and at least 3 technical replicates per condition. In parallel, cells were stained with WGA and DAPI and representative three-dimensional confocal images of each condition were taken with a CD7 using a Plan-Apochromat 20 × /0.7 objective and 1 × tubulense optovar by taking Z-stack images with a range between 10 and 15 µm. Images were processed using Zen Blue 3.2 software (Zeiss) and surface representation was done using Imaris 10.1.1.

### Zebrafish husbandry
Adults were raised under a 12/12 h light/dark cycle at the ZEFIX-CRBM zebrafish facility. Eggs were obtained by natural spawning and incu-bated at 28.5 °C in E3 medium (5 mM NaCl, 0.17 mM KCl, 0.33 mM CaCl$_2$, 0.33 mM MgSO$_4$). Zebrafish lines used in the study were: AB as wild- type and Tg(*mpeg1:mCherry-F*)$^{ump2Tg41}$.

### Zebrafish infection and treatment
Thirty hours post-fertilization (hpf) embryos were manually dechor-ionated using fine forceps and anaesthetized with 0.02% buffered MS222 (Tricaine; ethyl-3-aminobenzoate methanesulfonate salt). Microinjection was performed as previously described[80,81], except for inoculum preparation, which in this case was diluted in 2% Poly-vinylpyrrolidone (PVP) and supplemented with 0.05% phenol red for microinjection visualization. Embryos were injected with around 80

CFU of mScarlet-*Mmar* or 2% PVP (controls) in the caudal vein, rinsed, and transferred to small petri dishes (60 mm × 15 mm) containing E3. At 1 day-post-infection (dpi), embryos were anaesthetized and injected in the caudal vein with the different treatments: 2% PVP (control), MSN-AVA-TPP (10 and 20 ng), MSN-AVA-TPP@DOX (10 and 20 ng) and free DOX (1.2 pg and 500 pg). Embryos were then rinsed and transferred individually in 48-well plates for survival assays (n = 24 per condition per experiment), for CFU (n = 5 per time point per condition per experiment), and imaging (n = 10 per condition per experiment). For survival assays, embryos were monitored daily for 8 days and marked as dead in the absence of a heartbeat. Data are represented in a Kaplan-Meier graph. Bacterial burden from individual infected embryos was determined as previously described[80] at one- and three-days post-treatment (two- and four-days post-infection). Lysates were plated on 7H11 supplemented with 10% OADC and 50 μg/mL kanamycin and incubated for 7 days at 30 °C. Data are expressed as $\log_{10}$ CFU per embryo. At 3 dpi and/or 5 dpi, embryos received a second and/or third dose of 20 ng of MSN-AVA-TPP@DOX and were monitored for embryo survival as described above. Control embryos injected with the different treatments were monitored daily for the following symptoms: yolk opacification, edema, impaired equilibrium, impaired response to stimuli, bent body, abnormal blood flow, abnormal heartbeat, and default in swimming bladder inflation. Scores of 0 (absence), 1 (mild), and 2 (severe) were attributed to each criterion to generate a disease score ranging from 0 (normal healthy embryo) to 16 (dead embryo). Three biological experiments were performed.

### Zebrafish imaging
Embryos were anesthetized with 0.02% buffered MS222 and transferred individually in PhenoPlate™ 96-well microplates with black walls and cyclic olefin bottoms (Revvity) and imaged with a Cell Discoverer 7 (Zeiss). Treated embryos were imaged on a daily basis for 72 h to establish the clearance kinetics of MSN-AVA-TPP@DOX in vivo. First, whole embryo imaging was performed with a Plan-Apochromat 5x/0.35 objective and 0.5x tubulens optovar using the Tiles module. Then, 3D confocal images were generated with a Plan-Apochromat 20x/0.7 objective and 0.5x tubulens optovar by taking z-stack confocal images near the injection site with a range between 30 and 40 μm (2.6 μm slices). FITC signal was segmented to yield Fluorescent Pixel Counts (FPC) corresponding to the total surface of FITC signal per embryo ($\mu m^2$). To assess the localization of the MSN-AVA-TPP@DOX after treating infected embryos, whole embryos were imaged 48 h post-infection (24 h post-treatment) as described above, and 3D confocal images were acquired on infection loci with a Plan-Apochromat 20x/0.7 objective and 0.5x or 1x tubulens optovar by taking z-stack confocal images. All image analysis was performed using ZEN 3.2 software and Imaris 10.1.1.

### Statistical analysis
All analyses were performed using R and/or GraphPad Prism version 10.2.2 for Windows (GraphPad Software, CA, USA). In case of normal distribution, descriptive data are cited as mean ± SD and groups were compared using one-way or two-way ANOVA with Tukey's, Dunnett's or Šidák's multiple comparisons test. If data were not normal, descriptive data are cited as median and interquartile range and a non-parametric Kruskal-Wallis test with Dunn's multiple comparisons test was used to compare the groups.

### Reporting summary
Further information on research design is available in the Nature Portfolio Reporting Summary linked to this article.

## Data availability
Source data have been deposited on Figshare (https://figshare.com/s/8f2f6a4a2be6bd35da41).

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

## Acknowledgements

Figures were prepared using Inkscape, an open-source vector graphics software and BioRender. The authors are grateful to Chantal Cazevieille from Institute des Neurosciences Montpellier, INSERM, Université de Montpellier (Montpellier France) for her technical assistance with the SEM, Marc Plays from the ZEFIX platform and Claire Hamela for the zebrafish husbandry. We also thank the CEMIPAI (UAR 3725 CNRS Montpellier University) facility for the Cell Discoverer 7 microscope and the ZEN 3.2 software, Stephanie Viala and Myriam Boyer from the MRI BioCampus cytometry platform, MRI BioCampus for the Imaris 10.1.1 image analysis software. We thank Raphaelle Lopez (IRIM) for her valuable assistance with the myeloid cells and Dr Juan Luis Paris Fernández de la Puente from Instituto de Investigación Biomédica de Málaga y Plataforma en Nanomedicina (IBIMA Plataforma BIONAND) for his help with the analysis of the CLR release experiment. We also thank Estefanía García from the Mass Spectrometry facility from Universidad Complutense de Madrid for her assistance with the drug release experiments. This project was funded by the French National Research Agency grant 20-CE44-0019 (ILIome) and the Foundation pour la Recherche Médicale (Equipe FRM EQU202103012588) to L.K. M.G.-G. and M.V.-R. have been funded by the Comunidad de Madrid, by the Recovery, Transformation and Resilience Plan, and by Next Generation EU from the European Union (Ref: PR47/21-MAD2D-CM), and by the European Research Council (ERC-2015-AdG-694160, VERDI). FPB has been recipient of grants from the ANRS (AO2022-N°269560) and from the ANR (AAPG-Metaboden).

## Author contributions

J.J.A.C., Y.T., M.G.G., A.B., T.C., F.P.B. performed the experimental studies. J.J.A.C., Y.T., M.G.G., F.P.B., M.V.R., and L.K. carried out the analysis. J.J.A.C., Y.T., M.G.G., M.V.R., and L.K. wrote the manuscript. L.K. supervised the work.

## Competing interests

The authors declare no competing interests.
