## [Transparent Peer Review file · Nature Communications]

In vivo antimicrobial activity of engineered mesoporous silica nanoparticles targeting intracellular mycobacteria

Corresponding Author: Dr Laurent Kremer

Version 0:

Reviewer comments:

Reviewer #1

(Remarks to the Author)

In total this is a quite convincing paper and is suited for Nature Communications. There are however some issues which should be addressed.

1) Figure 1c. That 3 different NPs (even when there is only a surface modification) hat exactly the same mean value is highly unlikely. At least error bars should be given. And the dependence of the binning size should be discussed. In the present way this suggests an accuracy which is extremely unlikely. Even when measured several times the mean value of a 220 nm NP will vary a few nm. Same Figure, it should be written in which medium the measurements were performed. Most likely water. Additional data with a more realistic buffer (= salt) should be added. Why after the conjugation with -NH₂ there is such massive aggregation? When the authors explain this with the lower positive zeta potential this makes my above argument even more prominent: MSN-TPP has much more positive zeta potential that MSN-AVA2-TPP, and yet the meas size is the same with 1 nm precision, which would be hard to explain.

2) Figure 2a. The flow cytometry data look strange and and the graphs look cut. It is not likely that there are not a few scattered data points below the line parallel to the x-axis. These data should e shown. The authors explain their gating in Figure S1, still, the data should be un-cut and the gating can be indicated by a box as done
Apart for this this work is at very high level and nicely presented

Reviewer #2

(Remarks to the Author)

The manuscript describes the action of nanoparticles decorated with triphenylphosphonium and loaded with doxycyclin against mycobacterium marinum.

Although nanoparticles directed against mycobacteria have been described several times previously, these new particles seem to be particularly effective in binding mycobacteria and also show some activity in vivo. This is interesting for the field, although the in vivo treatment effects are moderate. We do see some delay in killing of the zebrafish, especially with repeated daily injections, but only a moderate reduction in bacterial levels in teh zebrafish.

It would have been interesting to see what the effect would be of loading the nanoparticles with more commonly used antibiotics for treating Mmarinum, such as clarithromycin or linezolid. Also testing the nanoparticles on other Mycobacteria would be interesting to show the broadness of this technique. The authors have published previously on M. abscessus and M avium....

The manuscript is well written, clear and the figures are beautiful and informative.

minor point: in line 160 the authors conclude there is no significant difference of the MSN beads by themselves, but later they do mention this difference as important, if it suits there reasoning... Perhaps mention in line 160 that the beads already do have an effect, even though it was (just) not significant.

Reviewer #3

(Remarks to the Author)

The communication showed the functionalization of mesoporous silica nanoparticles (MSN) to target the surface of M. marinum in vitro, as well as within infected macrophages. When the particles were loaded with doxycycline, the authors observed a pronounced decrease in the bacterial burden and hence they proposed the MSN nanosystems as a promising

alternative for the local treatment of *M. marinum* granulomas.

The authors have presented a well-written study, easy to follow and with impressive graphics. However, I believe the paper lacks some crucial findings necessary for acceptance in this high-impact journal. Therefore, I am requesting major revisions.

- Figure 7 summarizes the study, but the clinical translation is not adequately supported by the findings. Firstly, the authors describe *M. marinum* infections as cutaneous infections, making it unclear what type of administration route they envision. Only in the abstract is written "local treatment". Please clarify this along the text, indicating why type of administration is planned and the rationale behind for all the analysis. In particular, explain why Zebrafish were used as pre-clinical model by provide support for Zebrafish models in testing cutaneous infections.

- In case the authors planned for a local treatment, how they comment the choice of the medium used for the Mycobacteria-nanoparticle interactions? The Mycobacteria-nanoparticle interactions is strictly depending on the surface charges and this is affected by the pH of the medium and possibly the protein corona formation on the particles' surface. If the authors plan for IV injections, the serum could drastically alter the surface charge properties of the protein corona, thus affecting affinity. Therefore, the inclusion of serum in the study is necessary.

- Many experiments were conducted with only two technical replicates, which is a low standard for reproducibility.

- Fig 5. Why different concentration of DOX are used between the MSPs (125, 250 and 500 µg/mL) and the free drug samples (1.2 and 6 µg/mL)? Same for the Zebrafish treatment in Fig.6 ?

- The introduction missed a broad overview of the previous works on MSP for treating mycobacteria infections

Reviewer #4

(Remarks to the Author)

The biofilm inhibition results from the nanosystem are quite impressive compared to free DOX, and the same goes for its intracellular antibacterial activity. The in vivo findings are also very interesting and promising. That said, can these early in vivo results really indicate that the nanosystem would be well-tolerated in higher organisms, like mice, and eventually humans? Some clarification on this would be helpful.

Minor comments:

1) Introduction (Line 47): Remove the extra space in "100,000."

2) Why wasn't MSN-AVA-TPP@DOX tested to see if it blocks *M. marinum* uptake in MoDCs? Since this nanosystem seems to be the most promising for clinical applications, it would make sense to check if it can also prevent bacterial internalization.

3) Page 11, Line 296: It looks like there's a missing "2" after "AVA"—should this be "MSN-TPP and MSN-AVA2-TPP"? Worth double-checking.

Version 1:

Reviewer comments:

Reviewer #1

(Remarks to the Author)

The authors have improved the manuscript and it now can be accepted.

Reviewer #3

(Remarks to the Author)

I acknowledge the improvement of the manuscript and your rebuttal.

Reviewer #4

(Remarks to the Author)

The authors have addressed all of my suggestions.

Reviewer #1 (Remarks to the Author):

We would like to thank the reviewer for having devoted time to read, thoroughly evaluate and provide constructive criticism to our manuscript that will help to improve its overall quality.

In total this is a quite convincing paper and is suited for Nature Communications. There are however some issues which should be addressed.

1) Figure 1c. That 3 different NPs (even when there is only a surface modification) hat exactly the same mean value is highly unlikely. At least error bars should be given. And the dependence of the binning size should be discussed. In the present way this suggests an accuracy which is extremely unlikely. Even when measured several times the mean value of a 220 nm NP will vary a few nm. Same Figure, it should be written in which medium the measurements were performed. Most likely water. Additional data with a more realistic buffer (= salt) should be added. Why after the conjugation with -NH₂ there is such massive aggregation? When the authors explain this with the lower positive zeta potential this makes my above argument even more prominent: MSN-TPP has much more positive zeta potential that MSN-AVA₂-TPP, and yet the meas size is the same with 1 nm precision, which would be hard to explain.

We agree with the reviewer that the way in which data was shown was confusing. For that reason, Figure 1 has been redesigned. Former Figure 1c showed the size distribution of each group expressed in *number*. These graphs are generated by the DLS software following internal mathematical models. In this way, it is possible that groups of nanoparticles that are qualitatively different are represented in the same way. This is what happened here, and we would like to apologize for the misunderstanding. Instead, we have carried out new measurements and the results have been expressed in terms of mean size to show the differences among groups accurately (Figure 1c, plain colours). In addition, this way of expressing their behaviour aids to reflect the experimental fact that we observed that MSN-AVA-TPP behaved better, reason why it was chosen for subsequent experiments over MSN-TPP. Regarding the value observed for MSN-NH₂, it can be explained by the fact that nanoparticles are stabilized in aqueous media by electrostatic interactions. The existence of such interactions implies that nanoparticles are separated from each other by a certain distance. If the zeta potential is too low, the nanoparticles will not repel each other sufficiently, exceeding the minimum interaction distance, which translates into higher mean size values (aggregation). From an experimental point of view, the authors have traditionally found that such behaviour is observed for mesoporous silica nanoparticles when the zeta potential value falls in the range *ca.* -10 to +10 mV.

We also agree with the reviewer that the nanoparticles should be analysed in a more complex and realistic scenario. In this regard, rather than just using salt, we have employed Dulbecco's Modified Eagle Medium (DMEM) (Figure 1c, dashed colours). Culture medium contains both high salt concentration and proteins, providing such more realistic scenario. Overall, it could be observed that the mean size remained almost constant for MSN, MSN-TPP and MSN-AVA-TPP, regardless of the complexity of this solution. Exceptions could be observed for MSN-AVA₂-TPP and MSN-NH₂. While the former showed minor changes in this regard, the latter dramatically improved its behaviour. This can be explained by the formation of a protein layer around the nanoparticles that can improve interactions between nanoparticles, thereby avoiding the noted aggregation. We would like to highlight that the measurements performed for MSN-NH₂ are important in that they help to track the amino functionalization of plain MSN and the subsequent functionalization steps. Nonetheless, the findings of our work are unaffected by these values, since MSN-NH₂ does not conform the set of therapeutic candidates that have been biologically evaluated.

2) Figure 2a. The flow cytometry data look strange and the graphs look cut. It is not likely that there are not a few scattered data points below the line parallel to the x-axis. These data should be shown. The authors explain their gating in Figure S1, still, the data should be un-cut and the gating can be indicated by a box as done Apart for this this work is at very high level and nicely presented.

We appreciate the suggestions and agree with the reviewer. Because some of the bacterial strains express the fluorescent marker from an episomal plasmid, there is always a percentage of bacteria that lose the plasmid, leading to two bacterial populations detected by the cytometer (non-fluorescent and fluorescent). We chose to show only the population of mycobacteria expressing the fluorescent markers to avoid confusion between the strains, which is why the graphs look cut. To make the figure easier to read, we thus decided to change the representation of Fig. 2a from density plot to histogram. We hope this will satisfy the reviewer's demand.

Reviewer #2 (Remarks to the Author):

We would like to thank the reviewer for having devoted time to read, thoroughly evaluate and provide constructive criticism to our manuscript that will help to improve its overall quality.

The manuscript describes the action of nanoparticles decorated with triphenylphosphonium and loaded with doxycycline against mycobacterium marinum. Although nanoparticles directed against mycobacteria have been described several times previously, these new particles seem to be particularly effective in binding mycobacteria and also show some activity in vivo. This is interesting for the field, although the in vivo treatment effects are moderate. We do see some delay in killing of the zebrafish, especially with repeated daily injections, but only a moderate reduction in bacterial levels in the zebrafish.

We thank the reviewer for the suggestions and the constructive feedback. Regarding the first comment (moderate reduction in bacterial levels in the zebrafish), Fig. 6e only shows CFU counts from embryos after one dose of treatment (correlated with survival graph Fig. 6d), which is indeed a moderate effect. However, repeated daily injection leads to 89% of embryos survival. Since we have previously correlated embryos survival (Fig. 6d) with bacterial burden (Fig. 6e), and to reduce the number of embryos used in the study, we did not feel the need to recover CFUs from infected embryos treated with several doses, but have instead represented the bacterial burden with images (Fig. 6j), which shows that the reduction in bacterial levels is important and not moderate.

It would have been interesting to see what the effect would be of loading the nanoparticles with more commonly used antibiotics for treating M. marinum, such as clarithromycin or linezolid. Also testing the nanoparticles on other Mycobacteria would be interesting to show the broadness of this technique. The authors have published previously on *M. abscessus* and *M. avium*....

We appreciate the suggestion. We are unable to use our nanosystem with *M. abscessus* because the strain is resistant to doxycycline, and develops rapidly inducible resistance to clarithromycin (Richard *et al. Antimicrob. Agents Chemother.* 2020, 64(2): e01879-19). However, since we have previously described the use of zebrafish embryos to study *M. fortuitum* infections (Johansen and Kremer, *Front. Cell. Infect. Microbiol.* 2020, 10: 357; Roquet-Banères *et al. Antimicrob. Agents Chemother.* 2023, 67: e0160722), we have evaluated and confirmed the antibacterial activity of MSN-AVA-TPP@DOX against *M. fortuitum* *in vitro* (MIC/MBC/MBIC/MBEC), and the results were added in Supplementary Table 4. In addition, we have also tested this nanosystem *in vivo* in zebrafish embryos

infected with *M. fortuitum* and have validated the antibacterial activity with this other mycobacterial species (Supplementary Fig. 8).

Moreover, following the previous suggestion of the reviewer, we have loaded MSN-AVA-TPP with clarithromycin (MSN-AVA-TPP@CLR) and have shown that this nanosystem is active *in vitro* against *M. marinum* (MIC/MBC/MBIC/MBEC). Data were added to Supplementary Table 3.

The manuscript is well written, clear and the figures are beautiful and informative.

Minor point: in line 160 the authors conclude there is no significant difference of the MSN beads by themselves, but later they do mention this difference as important, if it suits there reasoning... Perhaps mention in line 160 that the beads already do have an effect, even though it was (just) not significant.

Unfunctionalized MSNs did not have a significant effect on bacterial internalization by phagocytes, as opposed to MSN-AVA-TPP (Fig. 3d-e). We later correlate these findings with the significant difference in bacterial burden in embryos treated with empty MSN-AVA-TPP (and not MSN) (Fig. 6e).

We hope that our modifications will meet the reviewer's expectations.

Reviewer #3 (Remarks to the Author):

We would like to thank the reviewer for having devoted time to read, thoroughly evaluate and provide constructive criticism to our manuscript that will help to improve its overall quality.

The communication showed the functionalization of mesoporous silica nanoparticles (MSN) to target the surface of *M. marinum* *in vitro*, as well as within infected macrophages. When the particles were loaded with doxycycline, the authors observed a pronounced decrease in the bacterial burden and hence they proposed the MSN nanosystems as a promising alternative for the local treatment of *M. marinum* granulomas.

The authors have presented a well-written study, easy to follow and with impressive graphics. However, I believe the paper lacks some crucial findings necessary for acceptance in this high-impact journal. Therefore, I am requesting major revisions.

1) Figure 7 summarizes the study, but the clinical translation is not adequately supported by the findings. Firstly, the authors describe *M. marinum* infections as cutaneous infections, making it unclear what type of administration route they envision. Only in the abstract is written "local treatment". Please clarify this along the text, indicating why type of administration is planned and the rationale behind for all the analysis. In particular, explain why Zebrafish were used as pre-clinical model by provide support for Zebrafish models in testing cutaneous infections.

We thank the reviewer for the suggestions and the constructive feedback. Regarding the administration route, because our nanosystem cannot be absorbed by the skin, topical treatment is not possible. In the case of human *M. marinum* infection, the granuloma is insulated and a good candidate for injection, which is why we consider injecting the treatment inside the injury. We have added in the conclusion that the type of administration will be local (Line 410). We have modified Figure 7 by removing the clinical translation to avoid any misinterpretation of our results.

Moreover, several studies have used the zebrafish to investigate skin infections (*Yersinia ruckeri* doi: [10.3390/biology10020166](https://doi.org/10.3390/biology10020166)), *Staphylococcus aureus* (doi: [10.3390/ph14060594](https://doi.org/10.3390/ph14060594)), *Aeromonas hydrophila* (<https://doi.org/10.3389/fmicb.2016.01219>), and most importantly *Mycobacterium marinum*-induced skin inflammation (<https://doi.org/10.3389/fimmu.2022.838425>). This study explored macrophage-dependent wound regeneration during mycobacterial infections, specifically with *M. marinum*. This research highlights the role of macrophages in tissue-specific immune responses and wound healing processes, offering insights into granulomatous inflammation relevant to human skin infections. These examples have been included in the Discussion section (lines 389-392).

We have chosen the zebrafish as a surrogate model to mimic the *M. marinum* granuloma formation in a natural host, and since our treatment can target the granuloma, we consider this as a good model. Granulomas formed in zebrafish embryos (commonly called pre-granulomas) are much less complex than granulomas in adult fish, mostly due to the absence of the adaptive immune response, resulting in a less compact structure. Therefore, it was not possible to inject the zebrafish granulomas locally because it completely disrupted the structure of the granuloma, which is why we chose to administer the treatment intravenously.

2) In case the authors planned for a local treatment, how they comment the chose of the medium used for the Mycobacteria-nanoparticle interactions? The Mycobacteria-nanoparticle interactions is strictly depending on the surface charges and this is affected by the pH of the medium and possibly the protein corona formation on the particles' surface. If the authors plan for IV injections, the serum could drastically alter the surface charge properties of the protein corona, thus affecting affinity. Therefore, the inclusion of serum in the study is necessary.

We thank the reviewer for raising this point. Although we do not plan for IV injections in humans, zebrafish were indeed treated intravenously, showing targeting of the infected area and overall survival of the embryos. Moreover, all *in cellulo* experiments were done in the presence of 10% FBS. Yet, it is true that the presence of a protein corona might affect the targeting ability. For that purpose, we have repeated the *M. marinum*-nanoparticle interaction using 10% human serum (HS) to verify if the serum would have an important effect. Although we saw a statistically significant decrease in the affinity between *M. marinum* and MSN-TPP as well as MSN-AVA-TPP when serum was added, the affinity still outperformed that of MSN. This proves that even in the presence of serum, the functionalization on the system with TPP increases the affinity towards the bacteria. These results were added in (loines 138-143) and Supplementary Fig. 2.

3) Many experiments were conducted with only two technical replicates, which is a low standard for reproducibility.

We apologize for this and have made modifications:

Fig. 2b) This was typo and was corrected (3 biological replicates instead of 2). The reason why only 2 technical replicates were done is because the number of events measured per technical replicate was very high (> 1 million events).

Fig. 4b) We have added a 4th biological replicate with 3 technical replicates (total of 9 technical replicates) to confirm reproducibility.

4) Fig. 5. Why different concentration of DOX are used between the MSPs (125, 250 and 500 µg/mL) and the free drug samples (1.2 and 6 µg/mL)? Same for the Zebrafish treatment in Fig.6?

We wanted to compare the effect of the nanoparticle's treatment with the effect of the free drug. In Fig. 4b, which is a biofilm experiment, we used 250 and 500 µg/mL of MSN-AVA-TPP@DOX, corresponding respectively to the MBIC and MBEC, and compared it to 6 and 12 µg/mL of free doxycycline. In Fig. 5b, we used 125, 250 and 500 of MSN-AVA-TPP@DOX, corresponding respectively to the MIC, MBC and twice the MBC, and compared it to 6. To stay consistent with our comparisons, we have now added 2 new concentrations of doxycycline; 3 µg/mL (to compare with 125 µg/mL of loaded nanoparticles) and 12 µg/mL (to compare with 500 µg/mL of loaded nanoparticles). In both experiments, we added the concentration of 1.2 µg/mL to mimic the concentration of DOX that reaches the skin after oral treatment with 1mg (Diehm *et al. PLoS One* **17**, e0270112 (2022)).

In the zebrafish experiments, in which we have used 10 and 20 ng of MSN-AVA-TPP@DOX and compared it to 500 pg of free doxycycline (amount release from 20 ng of MSN-AVA-TPP@DOX). We have removed the dose of 1.2 pg and have clarified in the text the rationale behind the values chosen (Lines 259-261).

5) The introduction missed a broad overview of the previous works on MSP for treating mycobacteria infections.

We agree with the reviewer's comment and added the following sentences with the corresponding reference: "Only a few examples of mycobactericidal MSNs are available to date, displaying nanocarriers capable of delivering a range of payloads from standard antimycobacterial drugs to novel antimycobacterial peptides²¹⁻²⁶." (line 97-100). Appropriate references have been added.

Reviewer #4 (Remarks to the Author):

We would like to thank the reviewer for having devoted time to read, thoroughly evaluate and provide constructive criticism to our manuscript that will help to improve its overall quality.

The biofilm inhibition results from the nanosystem are quite impressive compared to free DOX, and the same goes for its intracellular antibacterial activity. The in vivo findings are also very interesting and promising. That said, can these early in vivo results really indicate that the nanosystem would be well-tolerated in higher organisms, like mice, and eventually humans? Some clarification on this would be helpful.

Minor comments:

- 1) Introduction (Line 47): Remove the extra space in "100,000."

The extra space has been removed.

- 2) Why wasn't MSN-AVA-TPP@DOX tested to see if it blocks *M. marinum* uptake in MoDCs? Since this nanosystem seems to be the most promising for clinical applications, it would make sense to check if it can also prevent bacterial internalization.

We thank the reviewer for this suggestion. Our goal was to prove that the physical contact of the MSN-AVA-TPP with *M. marinum* influenced its internalization, without the presence of doxycycline which might also have an effect. But as suggested by the reviewer, since MSN-AVA-TPP@DOX would be used for clinical application, we have repeated the internalization

assay comparing the empty and loaded nanosystem (MSN-AVA-TPP vs MSN-AVA-TPP@DOX). We have shown that both nanosystems give similar infection patterns by preventing mycobacterial internalization (Fig 5.a-b).

- 3) Page 11, Line 296: It looks like there's a missing "2" after "AVA"—should this be "MSN-TPP and MSN-AVA₂-TPP"? Worth double-checking.

We have checked this, and we do mean MSN-AVA-TPP. As shown by Fig. 2b and Fig. 3b-c, MSN-TPP and MSN-AVA-TPP have similar targeting properties, whether it is against *Mmar* or against macrophages. MSN-AVA₂-TPP, however, was slightly less efficient.